# Cdk1-mediated DIAPH1 phosphorylation maintains metaphase cortical tension and inactivates the spindle assembly checkpoint at anaphase

Koutarou Nishimura[1], Yoshikazu Johmura[1,2], Katashi Deguchi[3], Zixian Jiang[3], Kazuhiko S.K. Uchida[4], Narumi Suzuki[2], Midori Shimada[5], Yoshie Chiba[2], Toru Hirota[4], Shige H. Yoshimura[3], Keiko Kono[1,6] & Makoto Nakanishi [1,2]

Animal cells undergo rapid rounding during mitosis, ensuring proper chromosome segregation, during which an outward rounding force abruptly increases upon prometaphase entry and is maintained at a constant level during metaphase. Initial cortical tension is generated by the actomyosin system to which both myosin motors and actin network architecture contribute. However, how cortical tension is maintained and its physiological significance remain unknown. We demonstrate here that Cdk1-mediated phosphorylation of DIAPH1 stably maintains cortical tension after rounding and inactivates the spindle assembly checkpoint (SAC). Cdk1 phosphorylates DIAPH1, preventing profilin1 binding to maintain cortical tension. Mutation of DIAPH1 phosphorylation sites promotes cortical F-actin accumulation, increases cortical tension, and delays anaphase onset due to SAC activation. Measurement of the intra-kinetochore length suggests that Cdk1-mediated cortex relaxation is indispensable for kinetochore stretching. We thus uncovered a previously unknown mechanism by which Cdk1 coordinates cortical tension maintenance and SAC inactivation at anaphase onset.

[1] Department of Cell Biology, Graduate School of Medical Sciences, Nagoya City University, 1 Kawasumi, Mizuho-cho, Mizuho-ku, Nagoya 467-8601, Japan.
[2] Division of Cancer Cell Biology, IMS, The University of Tokyo, 4-6-1 Shirokanedai, Minato-ku, Tokyo 108-8639, Japan. [3] Graduate School of Biostudies, Kyoto University, Yoshida-konoe, Sakyo-ku, Kyoto 606-8501, Japan. [4] Cancer Institute of the Japanese Foundation for Cancer Research (JFCR), Tokyo 135-8550, Japan. [5] Joint Faculty of Veterinary Medicine, Yamaguchi University, Yamaguchi 753-8515, Japan. [6] Okinawa Institute of Science and Technology Graduate University, 1919-1 Tancha, Onna, Okinawa 904-0495, Japan. These authors contributed equally: Koutarou Nishimura, Yoshikazu Johmura. Correspondence and requests for materials should be addressed to K.K. (email: KEIKO.KONO@OIST.JP) or to M.N. (email: mkt-naka@ims.u-tokyo.ac.jp)

During mitosis, animal cells undergo a dynamic reorganization of cell shape, in which cells become rounded to prepare for division in tissue layers[1–3]. Mitotic cell rounding is a complex process regulated by the fine-tuned coordination of multiple signaling events and is critical for chromosome segregation, development, tissue organization, and tumor suppression[4–9]. In order to generate the force for the spherical transformation, changes to the osmotic pressure[10] and the complete reorganization of the actin cytoskeleton[11–13] are required. The reorganization of the actin cytoskeleton is governed by at least three key modules: F-actin regulated by RhoA and an actin nucleator formin DIAPH1, Myosin II regulated by RhoA, Rac1, and Cdc42, and the Ezrin, Radixin, and Moesin family of proteins[2,12–16].

DIAPH1 is a member of the actin nucleator formin family of proteins. Proteins of this family are defined by their formin homology 1 (FH1) and formin homology 2 (FH2) domains. The formin homology 1 (FH1) domain is required for the interaction with the actin monomer-binding protein profilin, whereas the FH2 domain is responsible for actin filament nucleation[17]. Diaphanous-related formins (DRFs) comprise a subgroup activated by the binding of Rho-type small GTPases[18]. DRFs are involved in organizing various cytoskeletal structures such as filopodia, lamellipodia, and cytokinetic contractile rings. One of these, DIAPH1, is required for actin stress fiber formation[19] and maintenance of the cortical force during mitotic cell rounding[20].

The spindle assembly checkpoint (SAC) is a surveillance mechanism essential for faithful segregation of chromosomes. Activation of the SAC suppresses the anaphase-promoting complex/cyclosome (APC/C) in the presence of unattached and/or untensed kinetochore(s), thereby halting the metaphase to anaphase transition. Mechanisms underlying the prompt turning on and turning off of the SAC have been extensively studied in terms of the reversible phosphorylation of various substrates at the kinetochore by kinases and phosphatases[21]. However, the mechanistic link between the cortical tension during mitotic rounding and the SAC has been largely unexplored.

The increase in the cortex tension at prophase is triggered by Cdk1-dependent phosphorylation of Ect2[22], which in turn activates RhoA, leading to the accumulation of Rho-kinase-dependent myosin II[20] and DIAPH1-dependent F-actin on the cortex[14]. Thereafter, the cortex tension is maintained at a constant level during metaphase under the progressive accumulation of myosin II but with a decrease in actin thickness[14]. This is somewhat surprising since RhoA is activated at the cortex during early mitosis[23], raising the expectation that DIAPH1-dependent F-actin would progressively accumulate on the cortex and the tension would increase. Therefore, accumulation of F-actin by DIAPH1 on the cortex would be suppressed during metaphase independently of RhoA. In this study, we found that Cdk1 phosphorylated DIAPH1, which inhibited the interaction between DIAPH1 and profilin1 (PFN1) during metaphase. This inhibition is required for maintaining the cortical tension at a constant level and for the proper inactivation of the SAC at the onset of anaphase.

## Results

### Cyclin B1-Cdk1 phosphorylates the FH1 domain of DIAPH1.

RhoA-dependent DIAPH1 actin polymerization was activated at the onset of mitotic cell rounding. Subsequently, the cortex tension gradually increased and reached a maximum at pro/metaphase, but was maintained at a constant level during metaphase progression. Therefore, we speculated that the actin polymerization activity of DIAPH1 on the cortex would be negatively regulated during metaphase independently of RhoA. Thus, we

first examined the modification of DIAPH1 during mitosis. We detected an almost complete upward shift of bands, corresponding to 3×FLAG-DIAPH1 in HeLa cells, from mitotic shake-off at 30 and 60 min after RO-3306 release at which times prophase and metaphase cells were predominantly detected, indicating that the majority of 3×FLAG-DIAPH1 was post-transcriptionally modified in mitotic cells (Fig. 1a). A clear mobility shift of 3×FLAG-DIAPH1 bands was also detected in HeLa cells synchronized with nocodazole and was reversed with calf intestine alkaline phosphatase (CIP) (Fig. 1b), indicating that the mobility shift of DIAPH1 was due to its phosphorylation. The mobility shift was detected in RPE-1 cells as well as in floating U937 cells and HL60 cells (Supplementary Figure 1a). The mitotic upward shift of bands corresponding to wild-type (Wt) 3×FLAG-DIAPH1, ΔDAD, and ΔFH2/DAD was readily detected, but this shift was not detected in ΔFH1/FH2/DAD and ΔFH1 in HeLa cells (Fig. 1c). Although a band corresponding to ΔGBD/FH3 was not detected in the presence of nocodazole, the mitotic upward shift of ΔGBD/FH3 was detected in the presence of both nocodazole and MG-132, suggesting that ΔGBD/FH3 was degraded during mitosis by proteasomes (Supplementary Figure 1b). Consistent with this, an upward mitotic shift of FH1 was readily detectable (Fig. 2b). The FH1 domain proved to be difficult to analyze by mass spectrometry because only one trypsin cleavage site is present in the entire domain. This domain contains 17 serine and threonine residues including T607, S629, S640, and S665 that create an S/T-P motif, allowing us to determine whether these sites are phosphorylated in vivo using anti-MPM2 antibodies (Supplementary Figure 2a). IP-western blotting showed that DIAPH1 was phosphorylated at the S/T-P motif in nocodazole-treated mitotic but not interphase HeLa cells (Supplementary Figure 2b). As NetPhosK (http://www.cbs.dtu.dk/services/NetPhosK/) predicts that these residues are putative CDK phosphorylation sites, we examined whether cyclin B1-Cdk1 could phosphorylate GST-FH1 in vitro. Wt GST-FH1 and its T607A mutant were phosphorylated to a similar extent. Although the single mutants GST-S629A, GST-S640A and S665A were phosphorylated to a lesser extent, S629A/S640A/S665A(3A) and T607A/S4629A/S640A/S665A(4A) mutants were not phosphorylated by a purified Wt cyclin B1-Cdk1 (Fig. 1d). Taken together with the results from immunoblotting using phos-tag gels showing that other triple mutants, but not 3A, shifted upward when compared with the Wt (Supplementary Figure 2c), the results support the idea that the kinase responsible for mitotic DIAPH1 phosphorylation is cyclin B1-Cdk1 and that its phosphorylation sites are S629, S640, and S665.

To further confirm the mitotic phosphorylation of DIAPH1, we generated anti-phospho-DIAPH1 at S629 antibody (anti-pS629 antibody). The specificity of the antibody was confirmed by ELISA (Supplementary Figure 3a). This phospho-specific antibody recognized Wt 3×FLAG-DIAPH1 as well as its mutants (T607A, S640A, and S665A) in HeLa cells in the presence of nocodazole (Supplementary Figure 3b). The endogenous phosphorylation of DIAPH1 at S629 was first detected at 15 min and almost reached a plateau at 30 min after release from RO-3306 treated in HeLa cells (Fig. 1e), but not in cells simultaneously treated with RO-3306 and nocodazole (Fig. 1f), further supporting the idea that this process is catalyzed by cyclin B-Cdk1. Interestingly, activated RhoA and phosphorylation of MLC2 were detected within 5 min after release from RO-3306 treatment (Fig. 1e). Thus, the results suggest that activation of the Cdk1–Ect2–Rho axis occurs prior to Cdk1-mediated phosphorylation of DIAPH1 although RhoA activation was dispensable for Cdk1-mediated phosphorylation of DIAPH1 (Supplementary Figure 3c). Immunohistochemistry revealed that this phosphorylation was detected in prophase and metaphase HeLa cells with

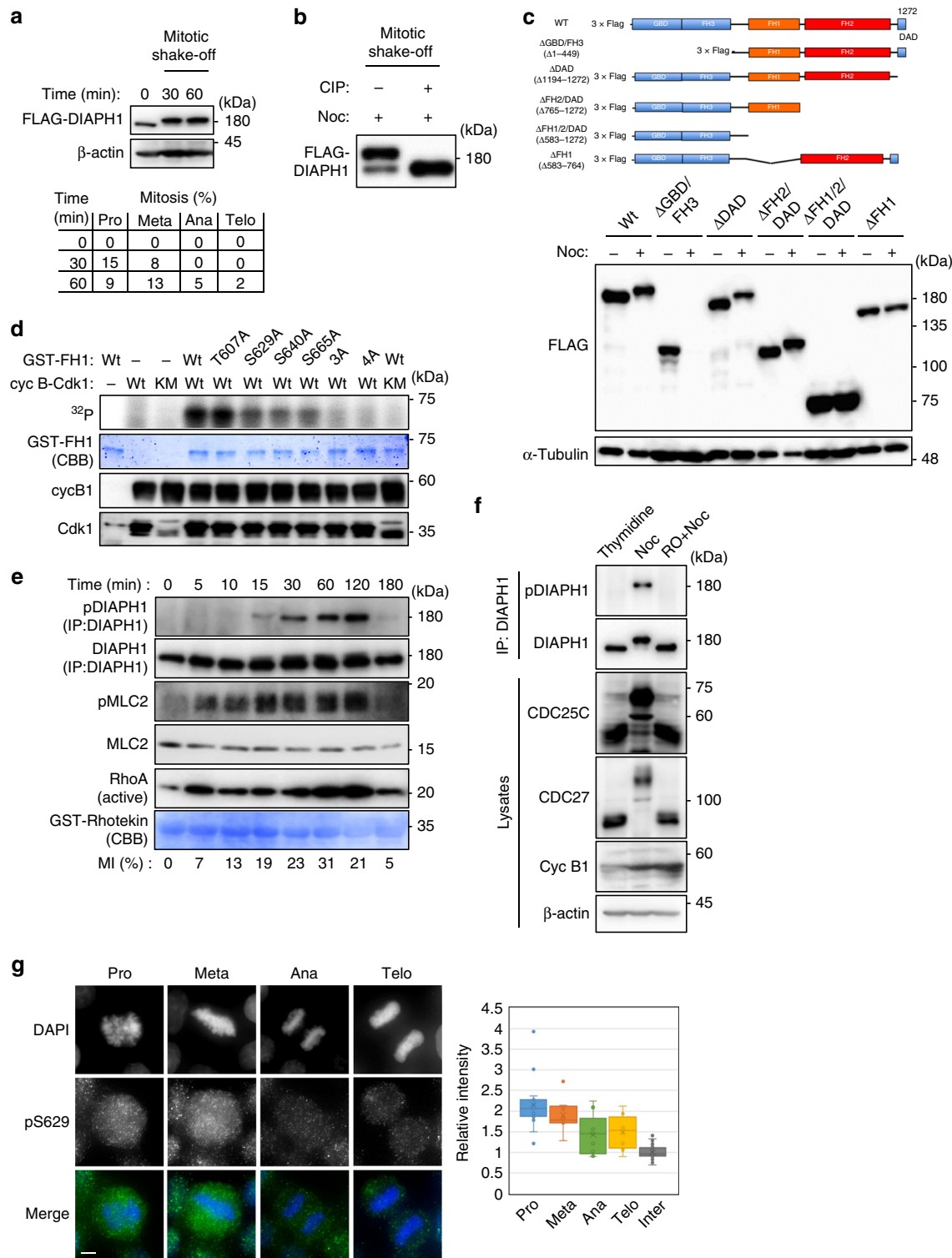

pan-cell staining excluding chromosomes (Fig. 1g). The phosphorylation signal was detected in HeLa cells synchronized with nocodazole, but was not detected in those synchronized with RO-3306 plus nocodazole (Supplementary Figure 3d). In addition, the phosphorylation signal was detected in inducible DIAPH1-knockdown HeLa (DIAPH1-KD) cells expressing Wt DIAPH1, but was not in those expressing 3A mutant (T607A/S640A/S665A) (Supplementary Figure 3e). These results further confirmed the specificity of the anti-pS629 antibody.

**The FH1 phosphorylation suppresses DIAPH1–PFN1 interaction.** Profilins mediate monomeric actin recruitment to formins[24,25]. GST-profilin1 (PFN1) bound preferentially to unphosphorylated DIAPH1 and only barely to the phosphorylated form while PFN1 consistently bound to actin in HeLa cells (Fig. 2a), suggesting that mitotic DIAPH1 phosphorylation likely downregulates its actin nucleation and elongation activity through the dissociation of PFN1. As expected, GST-PFN1 effectively bound to the Wt as well as to the FH1 domain of

**Fig. 1** Cyclin B1-Cdk1 phosphorylates the FH1 domain of DIAPH1. **a** Upper panels: HeLa cells expressing 3×FLAG-DIAPH1 were synchronized at G2 by RO-3306 and then released into fresh medium. Samples were collected by mitotic shake-off at the indicated times. Samples were subjected to SDS–PAGE and immunoblotting with the indicated antibodies. Lower panel: mitotic progression was monitored by DAPI staining and the mitotic cells (%) was determined as the percentages of the total cell number ($n = 100$). **b** 3×FLAG-DIAPH1 was immunoprecipitated from nocodazole-treated HeLa cells collected by mitotic shake-off, treated with CIP, and subjected to immunoblotting using anti-FLAG antibodies. **c** Upper panels: schema of DIAPH1 domain deletion mutants. Lower panels: HeLa cells expressing 3×FLAG-DIAPH1-Wt or its deletion mutants were treated with or without nocodazole. Cell lysates were subjected to immunoblotting using the indicated antibodies. **d** In vitro kinase assay ($^{32}$P) using purified Cyclin B1-Cdk1 Wt and its catalytically inactive KM mutant. GST-DIAPH1-FH1 (CBB), cyclin B1, and Cdk1 were immunoblotted using the indicated antibodies. **e** HeLa cells were treated as in **a**. Samples were collected at the indicated times and subjected to immunoprecipitation with anti-DIAPH1 antibodies or to pull-down experiments with purified GST-Rhotekin. The resultant precipitates were then subjected to immunoblotting with the indicated antibodies or to CBB staining. Mitotic cells were monitored by DAPI staining and the mitotic index (MI%) was determined as a percentage of the total cell number ($n = 100$). **f** Lysates from HeLa cells treated with a double-thymidine block, nocodazole shake-off (Noc), and RO-3306 treatment (10 min) plus nocodazole shake-off (RO + Noc) were subjected to immunoprecipitation with anti-DIAPH1 antibodies. The resultant precipitates (IP) and lysates were immunoblotted with the indicated antibodies. **g** Left: Asynchronous Hela cells were fixed with formaldehyde, and then subjected to immunostaining using anti-pS629 antibodies. Chromosomes were visualized by DAPI staining. Scale bar, 10 μm. Right: Relative signal intensities of cells in prophase, metaphase, anaphase, and telophase are shown. The background signal was subtracted in each field and the signal was then normalized using the average signal of three interphase cells. Whiskers are highest and lowest values excluding outliers. $n = 112$

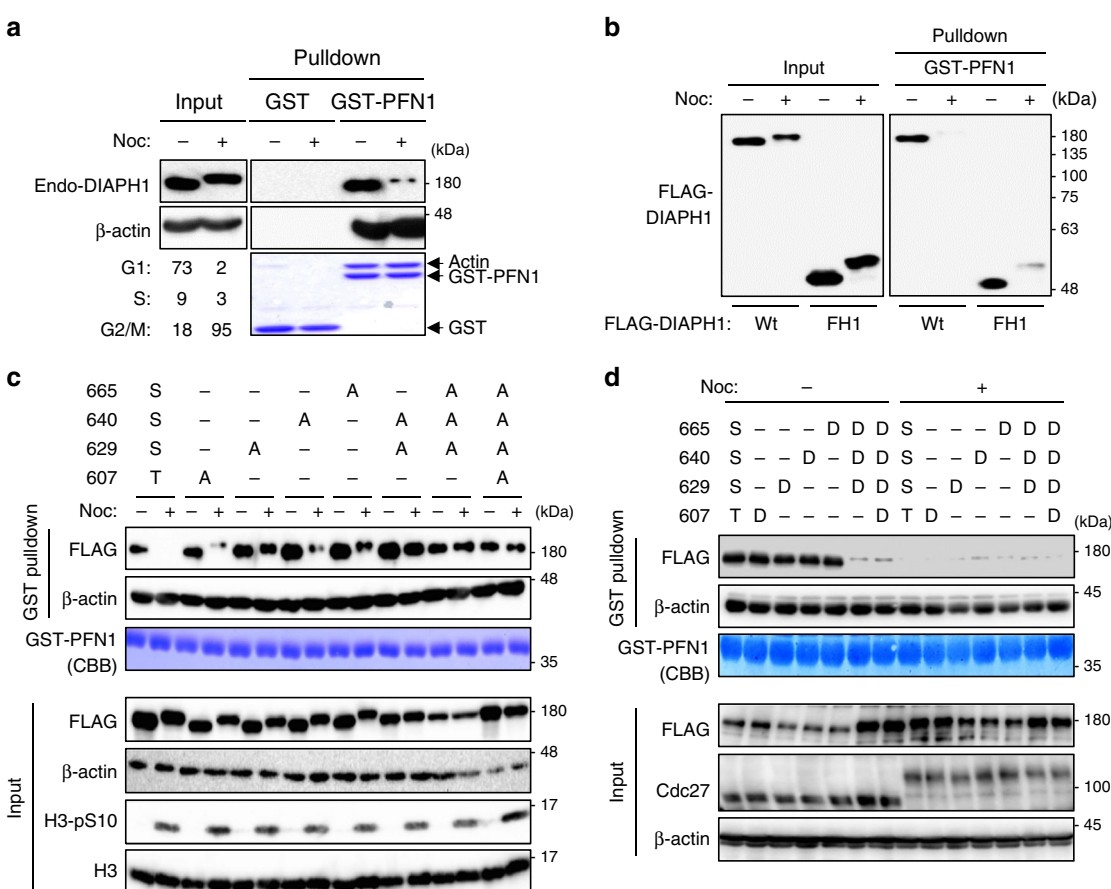

**Fig. 2** The FH1 phosphorylation suppresses DIAPH1–PFN1 interaction. **a** HeLa cells were treated with nocodazole for 18 h, and collected by mitotic shake-off. Cell lysates were incubated with GST-PFN1 and then subjected to a GST pull-down assay. The resultant precipitates were subjected to immunoblotting using the indicated antibodies. The DNA content of each sample was determined by FACS. **b** HeLa cells expressing 3×Flag-DIAPH1-Wt or its FH1 domain were treated and collected as in **a**. The lysates were incubated with GST-PFN1 and then subjected to a GST pull-down assay. The resultant precipitates were immunoblotted using anti-FLAG antibodies. **c**, **d** HeLa cells expressing 3×FLAG-DIAPH1-Wt or the indicated mutants were treated and collected as in **a**. Cell lysates were subjected to GST-pull-down experiments using GST-PFN1 and immunoblotting using the indicated antibodies. Transferred PVDF membranes were stained with Coomassie Brilliant Blue G-250 (CBB)

interphase but not mitotic HeLa cells (Fig. 2b). S629A and S665A slightly restored the binding to PFN1, and 3A or 4A mutants bound to PFN1 independent of nocodazole treatment in HeLa cells (Fig. 2c). Although the phospho-mimic mutants T607D, S629D, S640D, and S665D bound to PFN1 as effectively as the Wt did in the absence of nocodazole, 3D or 4D mutants failed to do

so (Fig. 2d), suggesting that phosphorylation of DIAPH1 at these three sites is sufficient for inhibition of DIAPH1 binding to PFN1.

**Expression of DIAPH1-3A enriches the cortical F-actin.** To determine the physiological significance of DIAPH1 mitotic

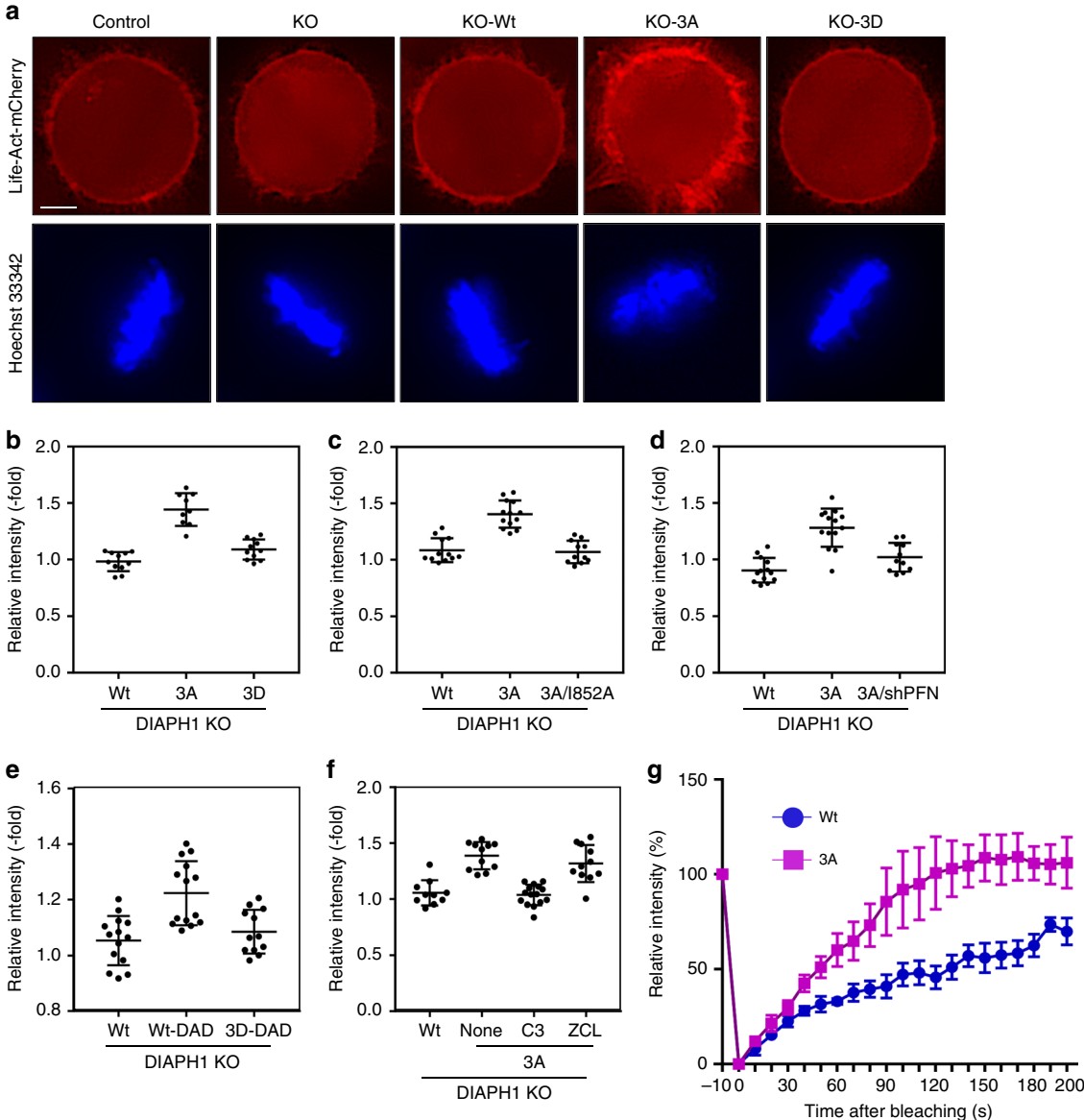

**Fig. 3** Expression of DIAPH1-3A accumulates F-actin on the cellular cortex. **a** Actin filaments are enriched on the cellular cortex of LifeAct-mCherry-KO-3A cells. Control or LifeAct-mCherry-KO-Wt, −3A, or −3D cells were stained with Hoechst 33342 for 10 min to monitor mitotic progression. Representative images shown are equally processed with BZ-II analysis software. Scar bar: 10 μm. **b** Cells as in **a** were imaged by microscopy without fixation. Relative signal intensities of mCherry at the cell cortex, cytoplasm, and background (the region without cells) were quantified using Image J software, the background was subtracted from each value, and their ratio (cortex,cytoplasm) was calculated. Wt: $n = 11$, 3A: $n = 9$, 3D: $n = 11$. Bars indicate means ± SD. **c** LifeAct-mCherry-KO-Wt, −3A, or −3A/I852A expressing-cells were analyzed as in **a**. Relative signal intensities of mCherry were quantified as in **b**. (Wt: $n = 12$, 3A: $n = 13$, 3A/I852A: $n = 11$). **d** LifeAct-mCherry-KO-Wt cells expressing tet-on shControl, or LifeAct-mCherry-KO-3A expressing tet-on shControl or tet-on shPFN1 in the presence of 1 μg/ml doxycycline were analyzed as in **a**. Relative signal intensities of mCherry were quantified in **b**. Wt: $n = 13$, 3A: $n = 14$, 3A-shPFN1: $n = 11$. **e** LifeAct-mCherry-KO-Wt, -Wt/DAD, or −3D/DAD cells were analyzed as in **a**. Relative signal intensities of mCherry were quantified in **b**. Wt: $n = 14$, Wt-DAD: $n = 14$, 3D-DAD: $n = 12$. **f** LifeAct-mCherry-KO-Wt or −3A cells were treated with or without 2 μg/ml C3 transferase or 50 μM ZCL278 for 30 min and were analyzed as in **a**. Relative signal intensities of mCherry were quantified in **b**. Wt: n = 10, 3A-none: $n = 11$, 3A-C3: $n = 15$, 3A-ZCL: $n$-11). **g** DIAPH1-KD-Wt or −3A cells were treated with SiR-Actin in order to visualize F-actin. Hoechst 33342 was added to the medium before the imaging. A circular area 0.1 μm in diameter at the actin cortex of a metaphase cell was bleached with the 633 nm laser (See Supplementary Figure 5). The relative signal intensity at the bleached region was quantified by Image J software. Bars indicate means ± SD. The signal intensities after bleaching were comparable to the cytosolic signal. Wt: $n = 6$, 3A: $n = 7$

phosphorylation, we stably expressed DIAPH1-3A at a level similar to that of endogenous DIAPH1 in stable DIAPH1-knockout HeLa (DIAPH1-KO) cells as well as in inducible DIAPH1-KD cells (Supplementary Figure 4a and 4b) expressing LifeAct-mCherry (LifeAct-mCherry-KD cells)[26], which monitors F-actin accumulation on the cortex. LifeAct-mCherry-KO cells expressing DIAPH1–3A (LifeAct-mCherry-KO-3A) showed a

significant enrichment of cortical F-actin at mitosis when compared with LifeAct-mCherry-KO-Wt or LifeAct-mCherry-KO-3D cells (Fig. 3a, b). These results are consistent with a previous report that DIAPH1 is essential for cortical F-actin enrichment[27]. This enrichment of cortical F-actin was not observed when we used LifeAct-mCherry-KO-Wt cells, and LifeAct-mCherry-KO cells expressing a 3A/I852A double-mutant lacking actin

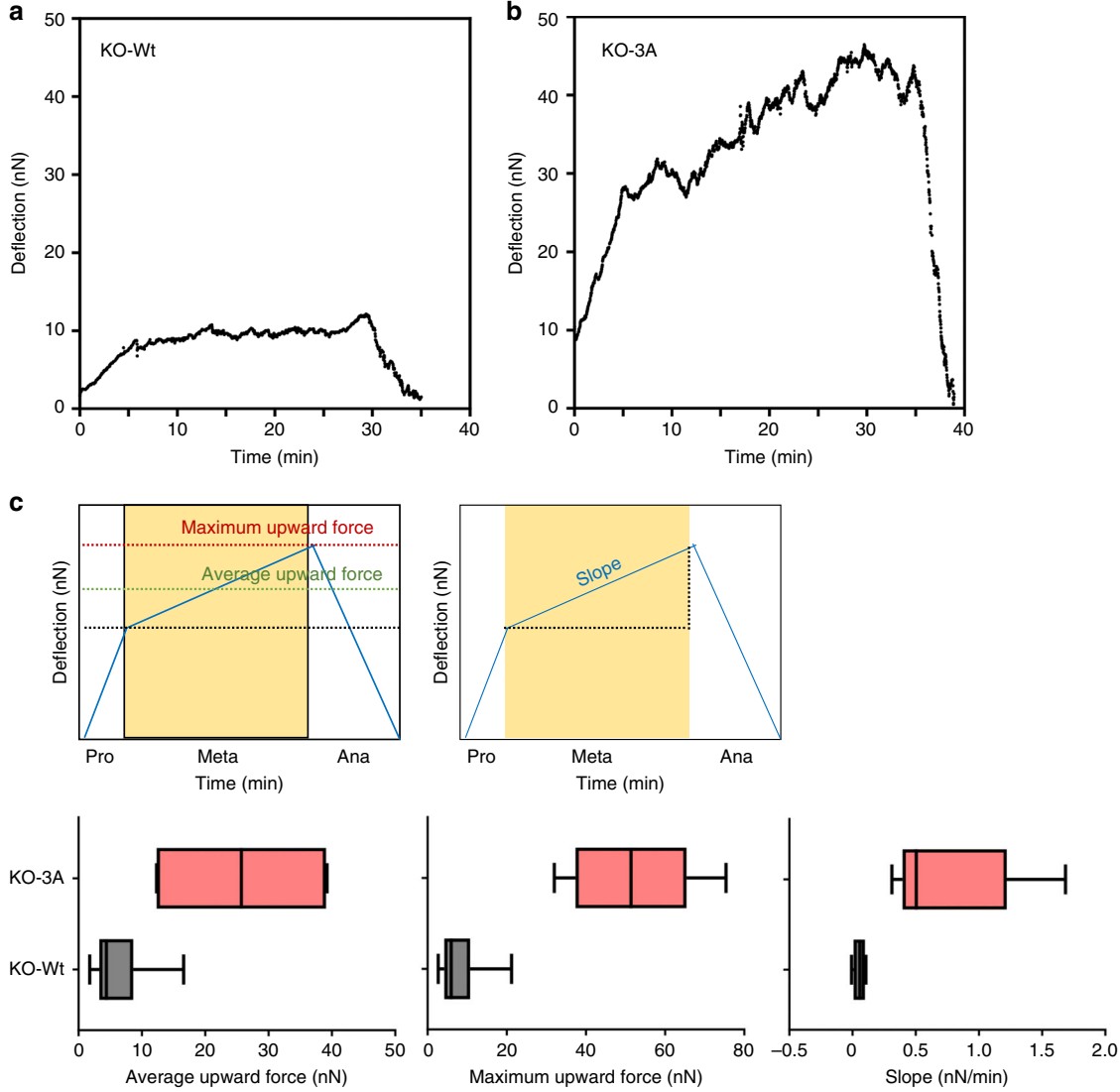

**Fig. 4** Expression of DIAPH1–3A increases cortical tension. Constant-height assay using AFM. Graphs show the upward deflection of the cantilever in the mitotic progression of KO-Wt (**a**) or −3A (**b**) cells. See Methods for the details. Representative results of six independent experiments are shown. **c** (Left) Average upward force during metaphase. (Middle) Maximum upward force during metaphase. (Right) Slope during metaphase. Box plots indicate median, interquartile values, and range. Cartoons (upper panels) illustrate the average upward force, maximum upward force, and slope

polymerization activity (Fig. 3c and Supplementary Figure 4c) or LifeAct-mCherry-KO-3A cells depleted of PFN1 (Fig. 3d and Supplementary Figure 4d). In addition, similar enrichment of cortical F-actin was observed in LifeAct-mCherry-KO cells expressing constitutively active DIAPH1 (ΔDAD), but this enrichment was suppressed when additional phospho-mimic mutations at T607/S640/S665 were introduced in an ΔDAD mutant (Fig. 3e and Supplementary Figure 4e). The enrichment of cortical F-actin in LifeAct-mCherry-KO-3A cells was inhibited by C3 transferase, a Rho inhibitor, but not by ZCL278, a Cdc42 inhibitor (Fig. 3f). This accumulation was also inhibited by depletion of PFN1 in LifeAct-mCherry-KO-3A cells (Fig. 3f). Prolonged mitosis by the treatment with MG-132 per se did not result in the enrichment of cortical F-actin (Supplementary Figure 5a). These results indicate that Cdk1-mediated phosphorylation of DIAPH1 suppresses an excess enrichment of cortical F-actin through inhibition of DIAPH1 actin polymerization activity. This inhibition of DIAPH1 activity is independent of Cdk1-mediated regulation of RhoA. In order to further examine the

cortical actin dynamics in DIAPH1-KD-3A cells, we performed fluorescence recovery after photobleaching (FRAP) analysis. With DIAPH1-KD-Wt cells labeled with SiR-Actin, the average half time of fluorescence recovery was ~120 s ($n = 6$), whereas it was ~45 s in DIAPH1-KD-3A cells ($n = 7$) (Fig. 3g and Supplementary Figure 6). These results suggested that cortical F-actin is more dynamic in DIAPH1-KD-3A cells than in DIAPH1-KD-Wt cells, consistent with our idea that the actin polymerization activity of DIAPH1-3A was enhanced.

Enrichment of cortical F-actin in LifeAct-mCherry-KO-3A cells raised the possibility of increased cortical tension. To monitor cortical tension, atomic force microscopy (AFM) and a constant-height assay[10,14] were employed. In H2B-mCherry-KO-Wt cells, the upward force increased at prometaphase and reached a plateau at metaphase, consistent with previous reports[10,14] (Fig. 4a). At anaphase onset, the force immediately decreased to the basal level. In contrast, the upward force was markedly high at prometa/metaphase and gradually increased during metaphase progression in H2B-mCherry-KO-3A cells

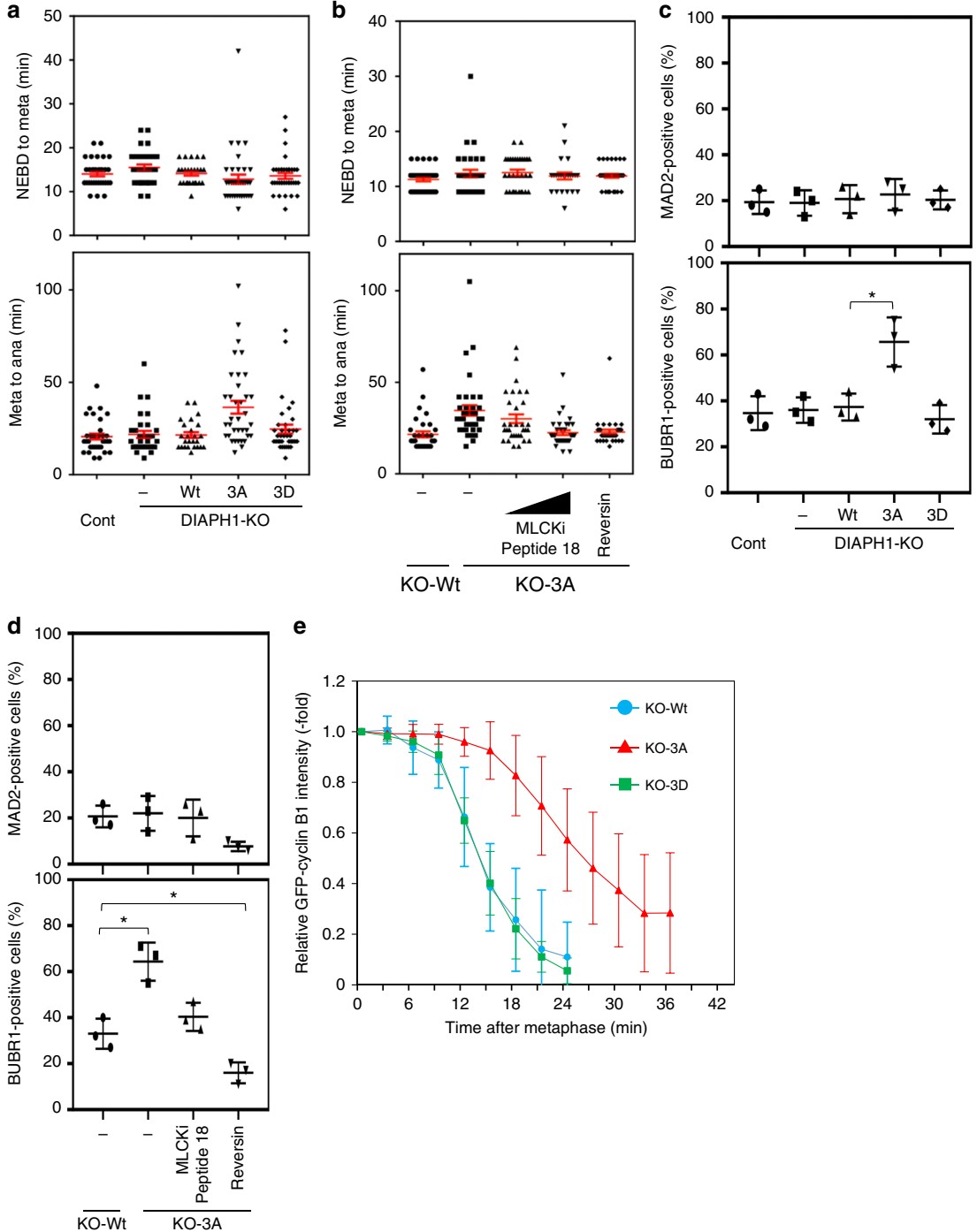

**Fig. 5** Proper anaphase onset requires DIAPH1 phosphorylation through inactivation of SAC. **a** Control (Cont) or H2B-mCherry-KO-Wt, −3A, or −3D cells were imaged by time-lapse microscopy every 3 min. The time interval between NEBD and metaphase or metaphase and anaphase was determined. Cont: $n = 33$, DIAPH1-KO: $n = 30$, KO-Wt: $n = 24$, KO-3A: $n = 34$, KO-3D: $n = 34$. Red bars indicate means ± SE. **b** H2B-mCherry-KO-Wt or −3A cells were treated with or without MLCK inhibitor peptide 18 (2 or 5 µM) or reversine (500 nM) for 30 min, and then imaged by time-lapse microscopy every 3 min. Wt: $n = 33$, 3A: $n = 35$ (2 µM MLCK inhibitor peptide 18) and $n = 30$ (5 µM MLCK inhibitor peptide 18), $n = 25$ (500 nM reversine). Durations were determined as in **a** and red bars indicate means ± SE. **c** Control (Cont) or KO-Wt, −3A, or −3D cells were stained with anti-MAD2 or anti-BUBR1 together with CREST antibodies. The number of metaphase cells showing co-localization of MAD2 or BUBR1 with CREST was counted (each $n = 200$). The results were obtained from three independent experiments. Bars indicate means ± SD. Statistical significance was determined by Student's $t$-test. $*p < 0.05$. **d** KO-Wt or −3A cells were treated with or without MLCK inhibitor peptide 18 (5 µM) or reversine (500 nM) for 30 min, and analyzed as in **a** (each $n = 200$). The results were obtained from three independent experiments and analyzed as in **c**. $*p < 0.05$. **e** GFP-cyclin B-KO-Wt, −3A, or −3D cells were analyzed by time-lapse microscopy every 3 min. Fluorescence intensities of cyclin B1-EGFP were measured during metaphase progression (each $n = 15$), and the mean ± SD was plotted for each time point. Levels of fluorescence were normalized to the fluorescent values obtained at the metaphase alignment

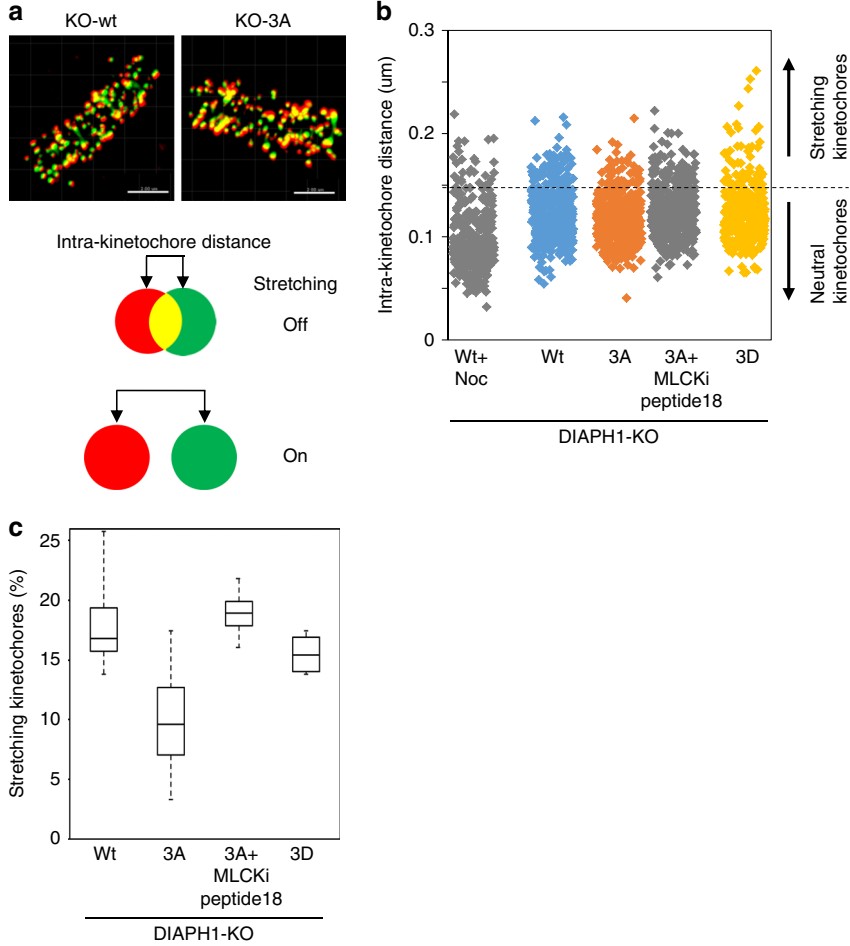

**Fig. 6** Impaired kinetochore stretching in KO-3A cells. **a** Cells were stained for Hec1 (red, outer kinetochore marker) and CREST (green, inner kinetochore marker), and metaphase cells were analyzed. Projections of three-dimensional images are shown. Bar: 2 μm. The intra-kinetochore distance is defined as the distance between the centroid of red and green signals, and kinetochores with a distance above the threshold (see below) were referred to as stretched kinetochores. **b** The intra-kinetochore distances of nocodazole (160 ng/ml (531.6 nM))-treated Wt HeLa cells (Wt + Noc., $n = 233$ kinetochores from 5 cells), and KO-Wt, −3A, −3A + MLCKi peptide 18 (5 μM) were plotted for 50 min, and KO-3D metaphase cells ($n = 318, 339,$ and 462 kinetochores from 4, 4, and 5 cells, respectively) were plotted as well. We determined the 95% confidence interval from the Wt + Noc experiment to be 0.156 μm, which was set as the threshold for stretching (dotted line, Uchida et al.[33]). **c** The rate of stretched kinetochores is shown in **b**. Box plots indicate median, interquartile values and range

(Fig. 4b). Maximum upward force as well as average upward force were higher in H2B-mCherry-KO-3A cells than that in H2B-mCherry-KO-Wt cells (Fig. 4c). Increased slope of upward force during metaphase was also detected in H2B-mCherry-KO-3A cells, but not in H2B-mCherry-KO-Wt cells. This higher and increasing upward force was not observed when anaphase onset was inhibited by MG-132 in H2B-mCherry-Wt cells, suggesting that the increase in cortical contractility in H2B-mCherry-KO-3A cells was not due to the extension of mitosis (Supplementary Figure 5b). Taken together, this indicates that Cdk1-dependent phosphorylation of DIAPH1 suppresses F-actin accumulation at the cortex, thereby maintaining cortical tension at a constant level during metaphase.

**SAC inactivation requires the cortical tension maintenance.** The maintenance of cortical tension at a constant level could be important for mitotic progression. In order to examine this possibility, we performed time-lapse imaging. We captured images every 3 min using H2B-mCherryl-KO-Wt, −3A, and −3D cells and determined the duration of the mitotic progression of

these cells. Nuclear envelop breakdown (NEBD) was recorded as the time at which all chromosomes were aligned at the metaphase plate, and anaphase was recorded as the time at which chromosomal separation was initiated to separate. The anaphase onset (the time from metaphase to anaphase) was significantly delayed in H2B-mCherry-KO-3A cells ($36.5 \pm 3.5$ min: means ± SE) compared with H2B-mCherry-KO-Wt cells ($21.5 \pm 1.5$ min) (Fig. 5a) although the rounding time and interval between NEBD to metaphase appeared comparable in these cells (Supplementary Figure 7a). H2B-mCherry-KO-3D cells showed kinetics of mitotic progression similar to that of H2B-mCherry-KO-Wt cells. A delay of anaphase onset was also observed in inducible H2B-mCherry-KD-3A cells (Supplementary Figure 7b). Importantly, treatment of H2B-mCherry-KO-3A cells with MLCKi peptide 18, which inhibits MLCK and consequently relaxes the cortical tension, reversed the delay in the anaphase onset to the control level in a dose-dependent manner (Fig. 5b), confirming that the delay was directly due to an increased cortical tension. The delay in anaphase onset in H2B-mCherry-KO-3A cells was also reversed when H2B-mCherry-KO-3A cells were treated with latrunculin, an inhibitor of actin polymerization (Supplementary Figure 7c),

further supporting the idea that this delay was due to enhanced actin polymerization by DIAPH1-3A.

We then examined whether the delay in the anaphase onset was due to prolonged activation of the SAC. Cells with activated SACs were evaluated by counting those positive for co-localization of MAD2, a marker of microtubule attachment, and BUBR1, a marker of tension, with CREST, a centromeric antigen[28,29]. KO-3A cells showed a significant increase in the population of BUBR1-positive cells when compared with KO-Wt or KO-3D cells whereas that of MAD2-positive cells appeared to be comparable in these cells (Fig. 5c, d). These results suggest that the increased cortical tension in KO-3A cells likely caused impaired intra-kinetochore tension, but not chromosome mis-alignment. The delayed anaphase onset by SAC activation in KO-3A cells was further confirmed by the finding that treatment of H2B-mCherry-KO-3A cells with a low dose of reversine[30], a specific inhibitor of the SAC, reversed the onset delay (Fig. 5b) as well as the decreasing population of cells positive for BUBR1 (Fig. 5d). We confirm that this low dose of reversine did not affect the chromosome biorientation. Treatment of KO-3A cells with MLCKi peptide 18 also reversed the increased population of cells positive for BUBR1, further confirming that SAC activation occurs because of the increased cortical tension (Fig. 5d). Impaired inactivation of SAC in KO-3A cells was further confirmed by analyzing the kinetics of cyclin B degradation, showing that reduction in the GFP-cyclin B intensity after metaphase was significantly delayed in GFP-cyclin B-KO-3A cells compared to that in GFP-cyclin B-KO-Wt or GFP-cyclin B-KO-3D cells (Fig. 5e). Taken together, these results suggest that Cdk1-mediated relaxing of the tension by suppression of F-actin accumulation regulates inactivation of the SAC at an appropriate time.

Previous reports showed that decreased cortical tension induces aberrant spindle assembly and chromosome misalignment[12,31]. We therefore examined the spindle architecture using GFP-tubulin as well as the behavior of chromosomes using living-cell imaging of H2B-mCherry-KO-3A cells during mitosis. Spindle architecture was apparently normal showing bipolar spindles (Supplementary Figures 8a and 8b). Chromosomal misalignment was not detected in H2B-mCherry-KD-3A cells (Supplementary Figures 8b, 8c, and Supplementary Movies 1 and 2). Lagging chromosomes and cytokinesis failure were comparable between H2B-mCherry-KD-Wt and H2B-mCherry-KD-3A cells (Supplementary Figure 8d), and changes in the diameter and spindle length were not apparent in these cells (Supplementary Figures 8e and 8f). Recently, kinetochore stretching was proposed as a requirement for SAC inactivation at anaphase onset[32,33]. Thus, we speculate that this mechanism might be involved in a delayed anaphase onset in KO-3A cells. Intra-kinetochore lengths of all chromosomes in metaphase cells were determined by measuring the distance between inner kinetochores stained with CREST and outer kinetochores stained with Hec1 (Fig. 6a). To interpret the obtained profile, we determined the threshold (0.156 μm) which included a 95% confidence interval for the intra-kinetochore length in the presence of nocodazole (in the absence of a spindle-pulling force) and then evaluated the longer intra-kinetochore distance of the stretched kinetochore (Fig. 6b). Interestingly, the population of stretched kinetochores were markedly lower in KO-3A cells compared with KO-Wt or KO-3D cells (Fig. 6c). These results indicate that kinetochore stretching is severely impaired in KO-3A cells. Importantly, this reduction in the population of stretched kinetochores in KO-3A cells was reversed when KO-3A cells were treated with 5 μM MLCKi peptide 18, suggesting that a high cortical tension at anaphase onset suppresses inactivation of the SAC by inhibiting sufficient kinetochore stretching.

## Discussion

Although actin polymerization by DIAPH1 is required for mitotic rounding[27], our current results clearly indicate that this DIAPH1 activity would be necessarily suppressed after rounding to prevent excess F-actin accumulation on the cortex, which would maintain the cortical tension at a constant level. This regulation is critical for inactivation of the SAC at the anaphase onset. At the onset of mitotic rounding, actin polymerization by DIAPH1 is activated by RhoA. This activation of DIAPH1 is mediated by Cdk1 through phosphorylation of Ect2, which in turn activates RhoA. Our results demonstrated that RhoA was immediately activated after release from RO-3306 treatment in HeLa cells whereas Cdk1-mediated phosphorylation of DIAPH1 reached a maximum at the metaphase to anaphase transition (Fig. 1e). Thus, at the onset of mitotic rounding, DIAPH1 is first activated by Cdk1–Ect2-dependent RhoA activation. During mitotic progression, Cdk1-mediated phosphorylation of DIAPH1 occurs gradually and likely serves as a negative regulator of RhoA-activated DIAPH1, which consequently suppresses actin polymerization at the cortex. This idea is further supported by results showing that the actin polymerization activity of constitutively active DIAPH1 independent of RhoA is still inhibited by this phosphorylation (Fig. 3e).

Phosphorylation-mediated inhibition of the actin polymerization activity of DIAPH1 appears to be critical for the prevention of excess accumulation of cortical F-actin. This notion is supported by results of a series of experiments carried out in the current study. (1) The phospho-mimic mutant 3D showed an accumulation of cortical F-actin similar to Wt DIAPH1 (Fig 3b). (2) A 3A/I852A mutant showed an accumulation of cortical F-actin similar to Wt DIAPH1 (Fig. 3c). (3) Depletion of PFN1 in LifeAct-mCherry-KO-3A cells reduced cortical F-actin to an amount similar to that of LifeAct-mCherry-Wt cells (Fig. 3d). (4) FRAP analysis revealed a rapid turnover of cortical F-actin in DIAPH1-KD-3A cells (Fig. 3g). Thus, the results suggest that the level of cortical F-actin has to be finely maintained by Cdk1-mediated positive and negative regulation of DIAPH1.

Although the level of the cortical upward force in H2B-mCherry-KO-3A cells appeared to be much higher than that in H2B-mCherry-KO-Wt cells, this only resulted in a modest delay of the onset of anaphase and a reduction of the cortical upward force to almost zero immediately after anaphase onset. These observations are likely explained by the fact that cortical contractility is determined by both cortical F-actin and myosin II; the former regulated by RhoA-DIAPH1, and the latter by RhoA-Rho-kinase axes. Therefore, after the onset of anaphase, reduced Cdk1 activity suppresses Rho-kinase activity, resulting in a release of myosin II from the cortex and a subsequent cortex relaxation, which leads to successful generation of the cleavage furrow in mitotic cells. In addition, DIAPH1 does not appear to be involved in the regulation of actomyosin ring formation because DIAPH1 depletion did not affect cytokinesis (Supplementary Figure 8d).

The mechanism by which cortical tension regulates the SAC is still an open question. A high cortical tension caused by an excessive F-actin accumulation may fail to correctly recruit and activate cortical dynein/dynactin, which generates spindle-pulling forces[34–36]. Although one would expect rigid cortical tension to increase astral microtubule pushing forces, an optimized level of cortical tension may be required for generation of maximal spindle-pulling forces, meaning that either a high or low cortical tension would decrease the forces. Alternatively, strong spindle-pulling forces may result in stable equilibrium of spindles[37], which would suppress kinetochore stretching. While the mechanism by which increased cortex tension by excess F-actin accumulation impairs kinetochore stretching requires further investigation, our results reveal a previously unknown function of

Cdk1 through which cortical tension is coupled with chromosome segregation.

## Methods

**Cell culture.** HEK293T (293T; ATCC) and HeLa (ATCC) cells, LifeAct-mCherry-HeLa Kyoto cells (provided by Dr. Cedric Cattin and Dr. Daniel J. Muller), Histone H2B-mCherry/Tubulin-LAP-HeLa Kyoto cells (also provided by Dr. Cedric Cattin and Dr. Daniel J. Muller) were cultured in Dulbecco's modified Eagle's medium (DMEM; Gibco) supplemented with antibiotics and 10% fetal bovine serum. To synchronize cells in M phase, they were treated with 100 ng/ml nocodazole (NDZ; Sigma) for 18 h, and mitotic cells were then isolated by a gentle shake-off. To synchronize cells in S phase, they were subjected to a double-thymidine block: cells were treated with 2.5 mM thymidine for 16 h, released, and at 8 h after the release, another round of thymidine treatment (2.5 mM thymidine for 16 h) was performed. For the MG-132 treatment, HeLa cells transduced with a 3×Flag-DIAPH1-ΔGBD/FH3 mutant were synchronized by nocodazole shake-off in the presence or absence of 20 μM MG-132 for 8 h. For live cell imaging during mitosis, 3×FLAG-DIAPH1 and 3xFLAG-DIAPH1(629A/640A/665A) were induced with 1 μg/ml doxycycline (Sigma-Aldrich) for 24 h prior to the imaging. Cell lines were not authenticated nor tested for mycoplasma contamination.

**shRNA or sgRNA primers.** shDIAPH1-shRNA Fw:
GATCCCCACCTGAAGGGACGGCTGGAACGTGTGCTGTCCGT
TCCAGCCGTCCCTTCAGGTTTTTTGGAAAT
shDIAPH1-shRNA Rv:
CTAGATTTCCAAAAAACCTGAAGGGACGGCTGGAACGGACAGCA-
CACGT TCCAGCCGTCCCTTCAGGTGGG
pX330-hDIAPH1-Fw:
CACCGCGGGACCCGGGACAAGAAGA
pX330-hDAIPH1-Rv:
AAACTCTTCTTGTCCCGGGTCCCGC
shPFN1-1-Fw:
GATCCCCGGCCAGAAATGTTCGGTGATTCTCGAGAATCACCGAA-
CATTTCTGGCCTTTTTGGAAAT
shPFN1-1-Rv:
CTAGATTTCCAAAAAGGCCAGAAATGTTCGGTGATTCTCGAGAAT-
CACCGAACATTTCTGGCCGGG
Human sgRNA DIAPH1-1 primer: 5′-TCTTCTTGTCCCGGGTCCCG-3′

**Plasmid construction.** DIAPH1 and DIAPH1 mutants (ΔGBD/FH3, ΔDAD, ΔFH2/DAD, ΔFH1/2/DAD, ΔFH1, FH1, 607A, 629A, 640A, 665A, 629A/640A, 3A [629A/640A/665A], 4A[607A/629A/640A/665A], 607D, 629D, 640D, 665D, 3D [629D/640D/665D], and 4D[607D/629D/640D/665D]) were generated using PCR and cloned into SalI and NotI sites of a pENTR-1A vector (Invitrogen) containing 3×Flag epitope. The resultant plasmid was mixed with CS-IV-TRE-RfA-UbC-Puro vector and reacted with Gateway LR clonase (Invitrogen) to generate the lentivirus plasmid. Lentivirus-based shRNA constructs and Tet-on-inducible lentivirus constructs were generated according to the previous report[38] as follows: to generate lentivirus-based DIAPH1-shRNA or PFN1-shRNA constructs, a 19–21 base shRNA-coding fragment (shDIAPH1: 5′-ACCTGAAGGGACGGCTGGA-3′) with a 5′-ACGTGTGCTGTCCGT-3′ loop or shPFN1-1: 5′-GGCCA-GAAATGTTCGGTGATT-3′) with a 5′-CTCGAG-3′ loop were introduced into pENTR4-H1 digested with AgeI/EcoRI using a set of shDIAPH1-shRNA Fw and shDIAPH1-shRNA Rv, or a set of shPFN1-1-shRNA Fw and shPFN1-1-Rv primers, respectively. To insert the H1tetOx1-shRNA into the lentivirus vector, we mixed the resulting pENTR4-H1-shRNA vector and CS-RfA-ETPuro vector with Gateway LR clonase (Invitrogen). RNAi-resistant DIAPH1-Wt, −3A, and −3D were cloned into XhoI and NotI sites of a CS-II-CMV-IRES2-Bsd vector. GST-PFN1 was generated using PCR and cloned into SalI and NotI sites of pGEX-6P-3 vector (GE). PCR was carried out using KOD-FX DNA polymerase (TOYOBO). All constructs were verified by DNA sequencing.

**Antibodies.** Polyclonal antibodies specific for a phosphorylated form of DIAPH1 at S629 were generated in rabbits with the keyhole limpet hemocyanin conjugated peptide CISpSPPSLP as an antigen. Antibodies used in this study are listed in Table S1.

**Lentiviral infection.** Lentiviruses expressing the respective genes were generated by cotransfection of 293T cells with pCMV-VSV-G-RSV-Rev, pCAG-HIVgp, and the respective CS-IV-TRE-RfA-UbC-Puro, CS-II-CMV-IRES-IRES-Bsd, CS-RfA-ETPuro using the calcium phosphate coprecipitation method. After 60 h, the lentiviral supernatant was collected and then used for infection of target cells. Cells infected with the indicated viruses were treated with 10 mg/ml of blasticidin (Invitrogen), 200 ng/ml of hygromycin (Sigma-Aldrich), and/or 2 mg/ml of puromycin (Sigma-Aldrich) for 2 to 3 days. Doxycycline (Sigma-Aldrich) was added to the medium at a concentration of 1 μg/ml for inducible expression of the respective genes.

**CRISPR/Cas9-mediated gene knockout.** A sgRNA for human DIAPH1 was ordered as an oligonucleotide, annealed, and cloned into the dual Cas9 and human sgRNA DIAPH1-1 expression vector pX330 (kindly provided by Dr. Feng Zhang) with a BbsI site. The plasmid (pX330-hDIAPH1-1) was transfected into HeLa cells using Lipofectamine 3000 (Invitrogen) according to the manufacturer's protocol. After 48 h incubation, the cells were split individually to make a clonal cell line. Deletion of DIAPH1 was confirmed by western blotting.

**Cell lysis and immunoblotting analysis.** Cells were lysed in TBSN buffer [20 mM Tris-Cl (pH 8.0), 150 mM NaCl, 0.5% Nonidet P-40, 5 mM EGTA, 1.5 mM EDTA, 0.5 mM $Na_3VO_4$, and 20 mM p-nitrophenylphosphate (PNPP)]. The resulting lysates were clarified by centrifugation at $15,000 \times g$ for 20 min at 4 °C, and the protein concentration was then measured by a bicinchoninic acid (BCA) assay (Sigma). Cell lysates were used at a protein concentration of 1 mg/ml. Proteins were separated by SDS-polyacrylamide gel electrophoresis (SDS–PAGE), transferred to a polyvinylidene difluoride (PVDF) (Immobilon-P; Millipore) membrane, and then detected by immunoblotting with the indicated antibodies using Luminate Forte Western HRP Substrate (Millipore). Antibody concentrations were as follows: Anti-Flag-M2 1:10,000 (Sigma), anti-α-Tubulin 1:10,000 (Sigma), Anti-beta-Actin 1:10,000 (Abcam) Anti-mDia1 1:1000 (BD Transduction Laboratories), Anti-phospho-Ser/Thr-Pro MPM2 1:1000 (Millipore), Anti-Glutathione S transferase (GST) 1:10,000 (Santa Cruz), Anti-H3, 1:10,000 (ab1791), and Anti-H3-pSer10 1:2000 (Millipore).

Mobility shift detection of phosphorylated DIAPH1 was performed with Phos-tag gels [5% polyacrylamide gels containing 7.5 μM Phos-tag (Wako) and 30 μM MnCl2]. After SDS–PAGE, Phos-tag gels were washed with transfer buffer (25 mM Tris, 192 mM glycine, 10% methanol) containing 1 mM EDTA for 10 min with gentle shaking and then with transfer buffer without EDTA for 10 min. All uncropped blots and gels are shown as Supplementary Figure 9.

**Immunoprecipitation.** Cell lysates were incubated with 1 μg of antibodies and 20 μl of Protein G Sepharose 4 Fast Flow beads (GE Healthcare) for 1 h at 4 °C. The beads were washed with TBSN buffer, resuspended in 2× Laemmli buffer, and then subjected to the SDS–PAGE and immunoblotting.

**Calf intestine phosphatase (CIP) treatment.** Cell lysates containing 10 μg of proteins were incubated with 10 U of CIP (NEB) and NEB buffer 3 at 37 °C for 120 min. The reaction was terminated by the addition of 5× SDS-sample buffer [3.125 mM Tris-Cl (pH 6.8), 25% 2-mercaptoethanol, 10% SDS, 25% sucrose, 25% BPB], followed by boiling for 5 min.

**CDK kinase assay.** Myc-Cdk1/Cyclin B1 and Myc-Cdk1 (kinase mutant)/Cyclin B1 was purified from insect (sf9) cells using anti-myc antibody. GST-DIAPH1-FH1-Wt and the mutants expressed in E. coli DH5α were induced by exposing cells to 1 mM IPTG for 1 h at 30 °C. GST fusion proteins were purified with Glutathione Sepharose 4B (GE Healthcare). Kinase reactions were performed according to the previous report[39] as follows: the kinase reaction were performed in a kinase buffer (50 mM Tris-Cl [pH 7.5], 10 mM MgCl2, 5 mM dithiothreitol, 2 mM EGTA, and 0.5 mM $Na_3VO_4$) supplemented with 5 μM ATP (10 μCi of [γ-$^{32}$P]ATP; 1 Ci = 37 Gbq) for 30 min at 30 °C, and then reactions were terminated by the addition of SDS-sample buffer, heated at 95 °C for 3 min, and then subjected to SDS–PAGE for autoradiography.

**GST-Rhotekin-RBD pull-down.** GTP-RhoA was quantitated according to the pull-down protocol provided with the Rho Activation Assay Biochem Kit. Briefly, the cells were harvested in the cell lysis buffer. The lysates (500 μg) were incubated with 30 μl of rhotekin-RBD beads for 1 h at 4 °C. Proteins bound to the beads were washed with the wash buffer, separated by SDS–PAGE, and analyzed by immunoblotting with anti-RhoA monoclonal antibody.

**Fluorescence cytochemistry.** Cells were plated on glass coverslips. After 48 h, the cells were fixed with 4% paraformaldehyde for 10 min at room temperature. For nuclear staining, the cells were incubated in PBS containing 0.1% Triton X-100 and 1 μg/ml 4′,6-diamino-2-phenylindole (DAPI; Sigma) for 30 min in the dark at room temperature. For phospho-DIAPH1(S629) staining, the cells were incubated with phospho-DIAPH1(S629) rabbit monoclonal antibody (1/250 dilution) at room temperature for 1 h. For MAD2, BUBR1, and CREST staining, cells were incubated with anti-MAD2 rabbit monoclonal antibody (1/100), anti-BUBR1 mouse monoclonal antibody (1/100), and anti-CREST human antibody (1/500) at the room temperature for 2 h. To visualize F-actin, LifeAct-mCherry HeLa Kyoto cells were cultured on a glass-based dish (Iwaki), placed on the stage of a BZ-9000 (Keyence) equipped with an environmental chamber (Keyence) which provided an adequate temperature, humidity, and CO2 control. Fluorescence images were acquired using DeltaVision Elite (GE Healthcare), AxioVision (Carl Zeiss), Cell-discoverer 7 (Carl Zeiss) or BZ-9000 (Keyence). FRAP analysis was performed using LSM-780 inverted confocal microscope (Carl Zeiss) equipped with a GaAsp Detector. Cells were incubated with 100 nM SiR-Actin (Cytoskeleton Inc.) for 1 h in order to visualize F-actin. Hoechst 33342 was added to the medium 5 min before

the imaging. A circular area of 1 μm diameter at the actin cortex of a metaphase cell was bleached with the 633 nm laser (100% laser power). The signal at the bleached region was quantified by Image J software. The signal intensities at time $-10$ s (before bleaching) and 0 s (immediately after bleaching) were set as 100% and 0%, respectively.

**Time-lapse microscopy**. Histone H2B-mCherry/Tubulin-LAP-HeLa Kyoto cells were cultured on a glass-based dish (Iwaki), placed on the stage of a BZ-9000 (Keyence) equipped with an environmental chamber (Keyence) which provided an adequate temperature, humidity, and $CO_2$ control. Time-lapse images were captured every 3 min for 16–24 h with a 536/40 emission filter. Images were analyzed using BZ-9000 software.

**Live cell imaging analysis of Cyclin B1-EGFP**. HeLa cells transfected with a bac plasmid expressing histone cyclin B1-EGFP were cultured on a glass-bottom dish (Matsunami, Japan) and stained with Hoechst 33342 for 10 min, placed on the stage of BZ-9000 fluorescent microscope (Keyence, Japan) equipped with an environmental chamber (KEYENCE, Japan) providing temperature, humidity, and $CO_2$ control. Time-lapse images were captured every 3 min. Fluorescence intensities of cyclin B1-EGFP in the cells were measured during metaphase with NIH imaging. Levels of fluorescence were normalized to the value obtained just after the metaphase alignment. metaphase was recorded as occurring from the time at which all chromosomes were aligned at the metaphase plate to the time at which chromosomal separation was initiated.

**FACS analysis**. Cells were fixed with 70% ethanol and stained with 20 μg/ml propidium iodide (PI; Sigma). Cell cycle profiles were analyzed using FACSVerse (BD Biosciences).

**GST-PFN pull-down**. GST-PFN1 was expressed in *E. coli* BL21-CodonPlus (Agilent Technologies) and purified with Glutathione Sepharose 4B (GE Healthcare). For the GST-PFN pull-down assay, 5 μg of bead-associated GST-PFN1 were incubated with 1 mg of cell lysate proteins for 1.5 h, and then precipitated. The precipitates were washed with TBSN buffer, and then subjected to immunoblotting.

**AFM analysis**. HeLa cells expressing histone H2B-mCherry were plated on a 35-mm-diameter glass-bottom dish (FD5040, World Precision Instruments, UK) 48 h before the experiment, and grown at 37 °C in a 5% $CO_2$ atmosphere. Cells were treated with doxycycline (Sigma-Aldrich, USA) at a final concentration of 10 μg/ml for 24 h before the measurement.

For the force measurement, an AFM scanning head (MFP-3D™, Asylum Research, USA) was mounted on an inverted microscope (IX71, Olympus, Japan) equipped with a temperature control system on the stage (Asylum Research, USA) and a fluorescence illumination/observation system was used. A tipless cantilever with a spring constant of ~0.2 N/m (TL-CONT, Nanosensors™, Switzerland) was used in all measurements. The spring constant of individual cantilevers was determined by a thermal method. The constant-height experiment was conducted according to the previous study[40] as follows: a glass-bottom dish containing HeLa cells in culture medium was placed on the microscope stage, and the temperature of the stage was maintained at 37 °C. A target cell, either in prophase or prometaphase, was chosen based on its chromosome morphology marked by an H2B-mCherry signal. The z-position of the cantilever was fixed at 8 μm from the glass surface above the target cell. Then, the deflection of the cantilever was recorded throughout mitosis. The deflection signal was converted to force using the following equation: $F = k \times d$, where $F$, $k$, and $d$ represent the force (nN), the spring constant of the cantilever (N/m), and the deflection (nm), respectively.

**Kinetochore stretching**. Measurement of the intra-kinetochore distance was based on the method of Uchida et al.[33]. Briefly, cells were fixed with 2% formaldehyde, and stained with anti-Hec1 (9G3, mouse monoclonal IgG, Novus), and the CREST antibody (in human serum) at 4 °C, overnight. The primary antibodies were probed with anti-mouse Alexa Fluor 568 (Invitrogen) and anti-human Alexa Fluor 488 (Invitrogen), respectively. Images were captured with an LSM880 equipped with an Airyscan module (objective lens: ×63 1.40 NA Plan Apochromat) (Carl Zeiss, Inc.). Images were analyzed with Imaris software ver. 7. 6. 4 (Bitplane Scientific Software) which automatically determines the $x$, $y$, and $z$ coordinates of the centroid of Hec1/CREST signals, to calculate the intra-kinetochore distances.

Kinetochore stretching is a phenomenon characterized by repetitive cycles of extension and recoiling of the kinetochore structure which emerges upon microtubule attachment. One experiment to estimate the degree of stretching is to quantitate how many kinetochores are detectably extended above a threshold level, which is defined by talking a 95% confidential interval of the intra-kinetochore distance in the absence of microtubule attachments, the condition created by nocodazole treatment. In the current experimental setup, a statistical calculation based on the data obtained with nocodazole treatment gave a threshold value of 0.16 μm. This allowed us to estimate that, in Wt cells, ~15% of kinetochores are extended and deformation of the remaining ~85% was below the threshold level, at any given time in unperturbed metaphase. Importantly, this does not mean that

only ~15% of kinetochores are statically stretched and the rest unstretched, but it does indicate that a majority of cellular kinetochores undergo a back and forth switching between extension and recoiling states, which allows inactivation of the spindle assembly checkpoint. A decrease in this rate has been shown to perturb SAC inactivation and to delay anaphase onset. Therefore, a decrease in the stretching rate to ~7% in the 3A mutant must be significant (5% is the minimum value, by definition) and thus sufficiently explains the mitotic delay.

**Statistical analysis**. Statistical analysis was performed using Student's $t$-test for independent variables. $P$-values $< 0.05$ were considered statistically significant.

## Data availability
All data are available from the corresponding authors upon reasonable request.

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

## Acknowledgements
We are grateful to Drs. C. Cattin, D.J. Muller, F. Zhang, A.A. Hyman, and E.K. Paluch for reagents, and C. Yamada, H. Takase, K. Yokote, S. Ohshima, and A. Maipas for technical assistance. A part of this study was conducted in the Core Laboratory, Nagoya City University Graduate School of Medical Sciences. This study was supported by MEXT/JSPS KAKENHI under Grant Numbers JP26250027, JP22118003, and JP16K15239, and by AMED under Grant Numbers JP17cm0106122, JP17fk0310111, and JP17gm5010001, as well as by Ono Medical Research Foundation, Princess Takamatsu Cancer Research Fund, and RELAY FOR LIFE JAPAN CANCER SOCIETY to M.N. K.K. was supported by MEXT/JSPS KAKENHI 15K19012, 17H04045, and JST-PRESTO JPMJPR1686.

## Author contributions
K.K. and M.N. conceived the idea of the project; K.N., Y.J., M.S., Z.J., K.S.K.U., S.H.Y., T.H., and K.K. planned the experiments; K.N., Y.J., K.D., Z.J., K.S.K.U., N.S., M.S., Y.C., and K.K. performed the experiments; K.N., Y.J., Z.J., K.S.K.U., S.H.Y., T.H., K.K., and M.N. analyzed the results; K.N., Y.J., Z.J., K.S.K.U., S.H.Y., and K.K. prepared the figures; and K.K. and M.N. wrote the manuscript with editing by all the other authors.

## Additional information

**Competing interests:** The authors declare no competing interests.

