## [Peer Review File · Nature Communications]

Reviewers' comments:

Reviewer #1 (Remarks to the Author):

In this manuscript, Nishimura et al. built up on previously described functions of the formin human mDia1/DIAPH1 in controlling cortical actin nucleation during mitotic cell division and propose an important inactivation step of DIAPH1 during metaphase which is mediated by Cdk1-dependent phosphorylation and critically involved in the progression into anaphase.

Overall, this represents an interesting study providing evidence for a novel mechanism of formin inactivation (mediated by phosphorylations within the FH1 domain) and proposing a new role of Cdk1 in tuning the dynamics of cortical tension to allow for mitotic division. However, there are several issues the authors should consider and address before publication. In the order of appearance in the manuscript:

- Given the critical role/function of DIAPH1 in the presented manuscript, I suggest to include it into the title.
- I suggest some editing of the abstract. Given the data shown, it's not clear to me whether the Cdk1-mediated suppression of F-actin accumulation is indeed required for stably maintaining cortical tension or whether the Cdk1-dependent phosphorylation represents a critical trigger for its relaxation. In line 41, write "relaxation" instead of "relaxing", in line 43 "and" instead of "with".
- Line 46, 47: Neither Ref. 12 nor Ref. 13 provide evidence for a DIAPH1-dependent cortical F-actin accumulation.
- Lines 46-54: I suggest to rephrase/rewrite this section to make it easier to follow and to better explain why the authors first started testing for SDS-PAGE migration behaviour of DIAPH1 during mitosis.
- Throughout the manuscript the authors should specify whenever they used stable cell lines and/or inducible expression systems.
- Throughout the figures, western blots need to indicate protein molecular weight (size)!! This is particularly important given the many different DIAPH1 truncations used and the repeated analysis of differences in mobility shifts.
- Figs. 1a-c appear pretty much redundant. The synchronization using RO-3306 (Fig. 1a) appears extremely inefficient (do only ~10% of cells divide after washout?) and the mobility shift is not convincing. Fig. 1b appears much more conclusive. The corresponding figure legend is confusing. It states "3xFLAG-DIAPH1 was immunoprecipitated from nocodazole (100 ng/ml)-treated HeLa cells as in (a)" although in (a) neither an immunoprecipitation nor nocodazole are mentioned. Were the cells in (c) also collected by shake off?
- Why is there no band in the deltaGBD/FH3 lane in Fig. 1d?
- The term "MI (%)" in Fig. 1f needs to be explained in the legend. How was this assessed and why is this different to Fig. 1a?
- Given the images in Fig. 1h, I wonder whether endogenous DIAPH1 shows cortical accumulation during mitotic rounding as could be expected by its proposed function. The specificity of the signal produced by the pS629 antibody should be further confirmed, e.g. using the knockout cells or phosphatase treatment.
- Line 102: delete "on"
- In Fig. 2a it should be specified whether endogenous DIAPH1 or Flag-DIAPH1 was studied. Do the numbers in Fig. 1a represent % of cells? Since Fig. 2a and 2b show similar experiments (only that Fig. 2b also includes the isolated FH1 domain), I suggest an equal labelling to make the figure more intuitive.
- Related to Fig. 3a and 3b and to overall verify that the effects of DIAPH1 or 3A-DIAPH1 are indeed mediated by an enhanced/increased actin assembly, the authors could in addition mutate a key actin polymerization residue within the FH2 domain of DIAPH1 (I845A in mouse mDia1).
- The legend for Fig. 3c states that representative examples are shown but, in contrast to the data shown in Fig. 4a (average time for the transition of metaphase to anaphase ~ 40min), the KO-3A cell

looks as if it doesn't reach anaphase at all.

- I suggest to include the information "average upward force" and "maximum upward force" into Fig.3d to make it more intuitive.
 - The statement in lines 129-131 should be weakened to e.g. "Taken together, this indicates that..."
 - How do the authors explain that KD-3A cells still manage to reach anaphase despite the loss of Cdk1-dependent phosphorylation and suppression of actin accumulation? In light of "only" a delayed onset of anaphase, the authors should tone down statements like "is essential" in line 156.
 - Line 136: remove "a"
 - Line 144: write "activated" instead of "activating"
 - For Fig.4b, it should be specified when exactly the drugs were applied.
 - Given the rather low number of cells analysed in Fig.4f and 4g, I wonder how the authors can ensure that the observed differences are not simply a result of slightly more or less advanced stages of metaphase.
 - Line 193, write "a" instead of "the".
 - Related to the Cdk1-dependent regulation of DIAPH1: The authors should discuss how they envisage a phosphorylation-mediated control of DIAPH1 activity in light of autoinhibition and the regulation by e.g. RhoA. To me, it would be interesting to see whether a constitutively active version of DIAPH1 (deltaDAD) could also be suppressed by Cdk1-dependent phosphorylation.
- minor:
- 1 B and C -how big is the phospho-shift?
all Western blots do not contain any protein size markers. this needs to be corrected.
- 4 A - are these rather moderate effects ? please comment.

Reviewer #2 (Remarks to the Author):

In this manuscript, Nishimura et al. use a combination of biochemistry, cell biology and biophysical approaches to uncover a direct link between Cdk1 and regulation of the actin cortex as cells pass through mitosis. The authors show that Cdk1 phosphorylates FH1 domain of DIAPH1 in mitotic cells, both in vitro and in vivo, and that this phosphorylation limits its association with Profilin and its ability to increase cortical tension, in a manner that can be counteracted by a Myosin inhibitor. Finally, they show that the expression of an active form of DIAPH1 that cannot be phosphorylated by Cdk1 delays the onset of anaphase, via the activation of the spindle assembly checkpoint. While they don't fully characterise the link between increased cortical tension and checkpoint activation, they suggest that Cdk1 may temper DIAPH1 activity to change kinetochore stretch to enable timely passage from metaphase into anaphase.

Since Cdk1-dependent regulation of myosin (via Ect2-Rho) has long been considered to be the dominant regulator of cortical tension in mitosis, this work represents a novel mechanism of mitotic tension regulation by Cdk1 through F-actin. Moreover, the biochemical data are excellent and very clear. However, the writing could be improved.

For example, in the results, the authors don't state up front which cell line they are using.

In addition, there are comments that the authors should consider addressing before publication.

General Comments that the authors should address:

1. There are a few puzzles that make the data hard to reconcile with previous work in the field.
 - i) How can cells balance Cdk1-Ect2-Rho induced cortical tension with Cdk1 DIAPH1 inhibition to regulate the cortex during passage through mitosis?
Is there something different about prophase-prometaphase and anaphase regulation of DIAPH1?
 - ii) What is the mechanism by which cortical tension inhibits anaphase onset? If forces are communicated from the cortex to the spindle, one would expect cortical tension to aid kinetochore

stretch. Explanations could include: the rigid cortex generated by activated DIAPH1 increases astral MT pushing forces, and/or DIAPH1 has a function within the spindle. Have the authors explored these possibilities?

iii) HeLa cells have been reported to exit mitosis with normal kinetics when actin is depolymerized using latrunculin. Is this true for 3A cells?

2. Have the authors tested whether DIAPH1 activity in the activated mutant depends on Rho activity and/or Cdc42 (Oceguera-Yanez 2005, Yasuda et al., 2004, Rosa et al., 2015)?

3. If, as the authors suggest, DIAPH1 phosphorylation reduces its association with Profilin, this should decrease actin filament assembly kinetics. It would be useful to look at cortical actin filament dynamics using FRAP. Does Profilin KD rescue the 3A actin phenotype? Also, do the DIAPH1 mutants affect actomyosin ring formation?

Specific points:

1. Figure 1A, there are no error bars. Also, the synchrony is very poor. Very few cells enter mitosis. Please show mitotic staging for 1B. It would also be good to state % anaphase in 1F.

2. Figure 1H: it would be good to see immunofluorescence microscopy examples of cells in all phases of mitosis (including at early stages of mitotic rounding), and in the presence of the Cdk1 inhibitor (RO-Noc).

3. Figure 1D. What happened in lane 4?

4. Figure 3A, show DNA stain as internal intensity control. Please add scale bars.

5. Figure 3C is very curious. It appears that cortical rigidity depends on time spent in metaphase. To test this, the plots should be blown up so one can directly compare the KO, KO-WT and KO-3A. It appears as if there is a steady increase in cortical contractility in the 3A cells during an extended mitosis. For comparison, one needs to see the same plot for WT cells that have been inhibited from exiting mitosis. The same is true for intensity – shown in 3A.

6. Figure 3D: please change the labels of the x-axes of the upper and middle plot to 'Average force (pN)' and 'Maximum force (pN)' respectively .

7. Figure 4A. The Figure should include rounding time.

8. Figure 4F and G: It would be good to see how the intra-kinetochore distance and the percentage of 'stretching kinetochores' changes with tension, by decreasing tension in a dose dependent manner with the MLCKi in KO-3A cells, since the authors made that point earlier with the anaphase onset and SAC positive cells measurements.

9. Please include confocal images of the spindle of the KO-3A cells, that were analysed for Extended Data Figure 4D.

Reviewer #3 (Remarks to the Author):

In this article the authors identify CDK1-dependent phosphorylation sites in DIAPH1 that inhibits its binding to profilin 1 and prevents excessive actin accumulation at the cortex during early phases of mitosis when the spindle checkpoint is active. This limits cortical tension, which the authors claim is important to allow proper inactivation of the SAC at anaphase.

In general, I found the mechanistic data on DIAPH1/PFN1 to be consistent with the authors conclusion, however, I feel that the phenotypic data that leads the authors to implicate a defect in SAC silencing was lacking and did not fully support the conclusions. My main issues are outlined below, along with suggestions for additional experiments that would strengthen the case for defective SAC silencing.

Major points

1) The authors mention that “chromosome misalignment was not detected in KO-3A cells” but show no data to substantiate these claims. Considering the cortical tension has been linked to chromosome misalignments before (ref 21,22), and any defects here would delay mitotic exit (as shown in fig 4) it is critical that the authors properly quantify chromosome alignment (preferably with fixed analysis and live H2B movies). It is not clear how NEB to meta analysis in 4A and 4B was performed – where these in H2B tagged cells? On a related note, it is surprising that reversine does not cause misalignment problems (i.e. increase NEBD-meta in b4) since this is well known to affect biorientation. Can the authors explain this? This make me worry that the assays used were not sensitive enough to detect alignment differences, which has serious implications for conclusions about and the SAC (which could be an indirect consequence of a subtle alignment defect).

2) It is not clear why pMCAK is chosen as a “SAC marker”. This is not a typical SAC marker and there are much better and more direct ones to use. I would suggest Mad2 as a marker a microtubule attachment and BUBR1 as a marker of tension. Both of these are well characterised SAC mediators that are needed for MCC assembly and are remove upon biorientation of kinetochore pairs. It would be important to quantify how Mad2/BubR1 change over time following microtubule attachment – is their loss from kinetochores delayed?

3) The quantification of intrakinetochore stretch in 4f/g is appropriate, but I cannot understand the seemingly arbitrary cutoff of approx 0.16uM to defined a “stretched” kinetochore. Looking at the data this would imply that the vast majority of WT cells are not “stretched” at a time when the SAC is silenced. How can this be explained? It looks to me that a line is drawn to make a point out of slightly more outliers in the WT/3D population. It is not possible to tell from 4f what percentage of kinetochores in the WT situation are above this cutoff, but it looks like the vast a large minority which is why I refer to these points as “outliers”. Although there may be variations in stretch as the kinetochore pairs oscillate, the vast majority should still be in a stretched state at metaphase (i.e. above the noco mean). In this case I expect the differences in the 3A would be small and unlikely to explain the mitotic exit differences. It looks to me that there are no real difference in the level of intrakinetochore stretch, but that there may be subtle difference in a small pool of hyperstretched kinetochores – would that be sufficient to explain the delayed exit. Do the authors conclude that a small amount of hyperstretched kinetochores are important to silence the SAC properly? I fail to see how these data substantiates the authors model.

There 3 major caveats above mean that I am not convinced that the clear mechanistic data actually causes a defect in SAC silencing. Another key experiment that could help to change that (in addition to the ones above): Cyclin B reporters to accurately quantify when Cyclin B degradation starts (i.e. a true marker of SAC inactivation) in relation to chromosome alignment, or better still, Mad2/BubR1 loss from the last KT.

In addition to the points above I have one major point regarding statistics. It is important that the authors clearly state the n numbers and what these refer to in the legends. This is critical because, for example, figure 4a and 4b show the critical increase in meta to ana duration in single cells, but it is unclear how many experiments this is from? It looks like the p values are calculated as if n = number of cells, which is not correct at all – a mean meta-ana duration for each experiment should be performed (i.e. means of “X” number of cells) then the stats calculated from those means based on n = number of experiments (for clarification see: Lazic et al, Plos Biol 2018). This should be clarified in the legends and applied to all figures panel throughout. Currently many of the panel do not state the n number.

Other minor points:

P9, L151 “decreasing” population of

Point-by-point responses to the reviewers' comments

Reviewer 1

- Given the critical role/function of DIAPH1 in the presented manuscript, I suggest to include it into the title.

According to the reviewer's suggestion, we changed the title to "Cdk1-mediated DIAPH1 phosphorylation maintains cortical tension during metaphase, which regulates inactivation of the spindle assembly checkpoint at anaphase onset" in the revised manuscript.

- I suggest some editing of the abstract. Given the data shown, it's not clear to me whether the Cdk1-mediated suppression of F-actin accumulation is indeed required for stably maintaining cortical tension or whether the Cdk1-dependent phosphorylation represents a critical trigger for its relaxation. In line 41, write "relaxation" instead of "relaxing", in line 43 "and" instead of "with".

According to the reviewer's comments, we revised the Abstract section as follows: "Cdk1-mediated phosphorylation of DIAPH1 regulates stable maintenance of cortical tension after rounding and inactivation of the spindle assembly checkpoint (SAC)." In the revised version, the words were changed to "relaxation" on line 41, to "and" on line 43. With respect to the reviewer's question, we performed FRAP analysis and found that the actin polymerization activity of a phosphorylation site mutant of DIAPH1 (DIAPH1-3A) is enhanced when compared to wild-type DIAPH1. In addition, Cdk1-dependent phosphorylation of DIAPH1 was detected during prophase to metaphase. Taken together, these results suggest that Cdk1-mediated suppression of F-actin accumulation is required for maintaining stable cortical tension during metaphase progression. We discussed these points in the revised version (page 11 line 19-page 12 line 5 and page 17 line 16-page 18 line 6).

- Line 46, 47: Neither Ref. 12 nor Ref. 13 provide evidence for a DIAPH1-dependent cortical F-actin accumulation.

We addressed these points in the revised version.

- Lines 46-54: I suggest to rephrase/rewrite this section to make it easier to follow and to better explain why the authors first started testing for SDS-PAGE migration behaviour of DIAPH1 during mitosis.

We added an introduction sentence in the revised version explaining the reason why we examined the SDS-PAGE migration of DIAPH1 during mitosis. (page 9 lines 9-15).

- Throughout the manuscript the authors should specify whenever they used stable cell lines and/or inducible expression systems.

Following the reviewer's advice, we addressed these points in the revised manuscript.

- Throughout the figures, western blots need to indicate protein molecular weight (size)!! This is particularly important given the many different DIAPH1 truncations used and the repeated analysis of differences in mobility shifts.

This is important. As advised by the reviewer, we added a protein molecular marker in all immunoblotting results in the revised version.

- Figs.1a-c appear pretty much redundant. The synchronization using RO-3306 (Fig. 1a) appears extremely inefficient (do only ~ 10% of cells divide after washout?) and the mobility shift is not convincing. Fig.1b appears much more conclusive.

As the reviewer indicated, Figs. 1a-c were redundant and the cell synchronization and mobility shift appeared to be inefficient and not convincing. Therefore, we eliminated Figure 1a in the revised version. Instead, we included mitotic staging in Figure 1b.

The corresponding figure legend is confusing. It states "3xFLAG-DIAPH1 was immunoprecipitated from nocodazole (100 ng/ml)-treated HeLa cells as in (a)" although in (a) neither an immunoprecipitation nor nocodazole are mentioned. Were the cells in (c) also collected by shake off?

We improved the legends for the revised Figure 1a and 1b.

- Why is there no band in the deltaGBD/FH3 lane in Fig.1d?

This point is important. Although a band corresponding to Δ GBD/FH3 was not detected in the absence of MG132, it was readily detectable in the presence of MG132, showing a clear mitotic upward shift. Thus, the results suggest that Δ GBD/FH3 is a very unstable protein that is degraded by proteasomes during mitosis. These points were discussed in the revised text and the results are shown in the revised Supplementary Figure 1b.

- The term "MI (%)" in Fig.1f needs to be explained in the legend. How was this assessed and why is this different to Fig.1a?

We described the term "MI(%)" and how it was determined in the revised legend for Figure 1g.

- Given the images in Fig.1h, I wonder whether endogenous DIAPH1 shows cortical accumulation during mitotic rounding as could be expected by its proposed function. The specificity of the signal produced by the pS629 antibody should be further confirmed, e.g. using the knockout cells or phosphatase treatment.

With respect to the localization of DIAPH1, it was reported that DIAPH1 (mDia1) localizes to cytoplasm (Kosako H et al. Oncogene 2000). However, the actin polymerization activity of DIAPH1 requires activated RhoA, and activated RhoA was reported to be predominantly enriched at the cortex during early to middle mitosis (Yoshizaki H et al. JCB 2003). Therefore, the results suggest that the actin polymerization activity of DIAPH1 could be enhanced predominantly at the cortex during mitosis. As the reviewer advised, additional experiments further confirmed the specificity of the signal produced by the pS629 antibody. The phosphorylation signal was detected in cells synchronized with nocodazole, but was markedly attenuated in those synchronized with RO-3306 plus nocodazole (Supplementary Figure 3c). In addition, the phosphorylation signal was detected in inducible DIAPH1-knockdown HeLa (DIAPH1-KD) cells expressing Wt DIAPH1, but was not in those expressing 3A mutant (T607A/S640A/S665A). The results are shown in the revised Supplementary Figures 3c and 3d and discussed in the text (page 9 lines 5-11).

- Line 102: delete “on”

We removed the “on” in the revised version.

- In Fig.2a it should be specified whether endogenous DIAPH1 or Flag-DIAPH1 was studied. Do the numbers in Fig.1a represent % of cells? Since Fig.2a and 2b show similar experiments (only that Fig.2b also includes the isolated FH1 domain), I suggest an equal labelling to make the figure more intuitive.

According to the suggestion, we specified that endogenous DIAPH1 was analyzed in the revised Figure 2a. We revised Figure 2b to make the figure more intuitive, and changed labelling to correspond to that of Figure 2a.

- Related to Fig.3a and 3b and to overall verify that the effects of DIAPH1 or 3A-DIAPH1 are indeed mediated by an enhanced/increased actin assembly, the authors could in addition mutate a key actin polymerization residue within the FH2 domain of DIAPH1 (I845A in mouse mDia1).

This point is very important and well taken. According to the suggestion, we examined the enrichment of F-actin at the cortex in KO-3A and KO-3A/I852A cells and found the

suppression of enrichment of F-actin at the cortex in KO-3A/I852 cells. The results are shown in the revised Figure 3c and discussed in the revised text (page 11 lines 2-5).

- The legend for Fig.3c states that representative examples are shown but, in contrast to the data shown in Fig.4a (average time for the transition of metaphase to anaphase \sim 40min), the KO-3A cell looks as if it doesn't reach anaphase at all.

This point is also important. We showed more representative examples in the revised Figures 4a and 4b.

- I suggest to include the information “average upward force” and “maximum upward force” into Fig.3d to make it more intuitive.

According to the suggestion, we have included cartoons providing information on the “average upward force” and “maximum upward force” into the revised Figure 4c.

- The statement in lines 129-131 should be weakened to e.g. “Taken together, this indicates that...”

According to the comment, we revised the sentence as follows: “Taken together, this indicates --“.

- How do the authors explain that KD-3A cells still manage to reach anaphase despite the loss of Cdk1-dependent phosphorylation and suppression of actin accumulation? In light of “only” a delayed onset of anaphase, the authors should tone down statements like “is essential” in line 156.

This point is very important and discussed in the revised version. Cortical contractility is determined by both cortical F-actin and myosin II; the former regulated by RhoA-DIAPH1, and the latter by RhoA-Rho kinase axes. Therefore, after onset of anaphase, reduced Cdk1 activity suppresses Rho kinase activity, resulting in a release of myosin II from the cortex and subsequently a cortex relaxation, which leads to success in generating the cleavage furrow in mitotic cells. In addition, DIAPH1 does not appear to be involved in the regulation of actomyosin ring formation because DIAPH1 depletion did not affect cytokinesis. These observations likely explain that KO-3A and KD-3A cells could manage to reach anaphase and showed only a modest delay in the anaphase onset. These points are discussed in the revised manuscript (page 18 lines 7-18).

- Line 136: remove “a”

According to the indication, we removed “a” in the revised version.

- Line 144: write “activated” instead of “activating”

According to the indication, we replaced “activating” with “activated” in the revised version.

- For Fig.4b, it should be specified when exactly the drugs were applied.

We specified the drugs used in the revised Figure 5b.

- Given the rather low number of cells analysed in Fig.4f and 4g, I wonder how the authors can ensure that the observed differences are not simply a result of slightly more or less advanced stages of metaphase.

This point is important. To address the question of whether the kinetics of kinetochore stretching might change during metaphase, we filmed mitotic HeLa cells expressing EGFP-CENP-A/mCherry-Mis12 through the metaphase-to-anaphase transition, and cells in early metaphase (20-12 min before the anaphase onset) and in late metaphase (8-0 min before the anaphase onset) were analyzed for their intra-kinetochore status as we previously reported (Uchida et al., 2009). We found that neither the frequency nor duration of kinetochore stretching changed significantly between the early and late metaphase datasets (please see the figure shown below), which discounts the possibility that the differences we observed in Wt and 3A mutant cells in the revised Figures 7b and 7c reflects a difference in metaphase stages.

18 cells were filmed in total (6 time-lapse imaging)
early (n=20) and late (n=36) metaphases were analyzed

- Line 193, write “a” instead of “the”

We changed “the” to “a” in the revised version.

- Related to the Cdk1-dependent regulation of DIAPH1: The authors should discuss how they envisage a phosphorylation-mediated control of DIAPH1 activity in light of autoinhibition and the regulation by e.g. RhoA. To me, it would be interesting to see whether a constitutively active version of DIAPH1 (deltaDAD) could also be suppressed

by Cdk1-dependent phosphorylation.

This point is very important and well taken. At the onset of mitotic rounding, actin polymerization by DIAPH1 is activated by RhoA. This activation of DIAPH1 is mediated by Cdk1 through phosphorylation of Ect2, which in turn activates RhoA. Our results demonstrated that RhoA was immediately activated after release from RO-3306 treatment in HeLa cells whereas Cdk1-mediated phosphorylation of DIAPH1 reached a maximum at the metaphase to anaphase transition (Figure 1e). Thus, at the onset of mitotic rounding, DIAPH1 is first activated by Cdk1-Ect2-dependent RhoA activation. During mitotic progression, Cdk1-mediated phosphorylation of DIAPH1 occurs gradually and likely serves as a negative regulator of RhoA-activated DIAPH1, which consequently suppresses actin polymerization at the cortex. This idea is further supported by the results showing that the actin polymerization activity of constitutively active DIAPH1 independent of RhoA is still inhibited by this phosphorylation (Figure 3e). The results are shown in the revised Figures 1e and 3e, and discussed in the text (page 11 lines 6-10 and page 16 line 18-page 17 line 15).

minor:

1 B and C -how big is the phospho-shift?

all Western blots do not contain any protein size markers. this needs to be corrected.

This is important. According to the indication, we added a protein molecular marker in all immunoblotting results in the revised version.

4 A - are these rather moderate effects ? please comment.

Please see the above response to your question.

Reviewer 2

1. There are a few puzzles that make the data hard to reconcile with previous work in the field.

i) How can cells balance Cdk1-Ect2-Rho induced cortical tension with Cdk1 DIAPH1 inhibition to regulate the cortex during passage through mitosis?

Is there something different about prophase-prometaphase and anaphase regulation of DIAPH1?

This comment is almost the same as that of reviewer 1. This point is very important and well taken. At the onset of mitotic rounding, actin polymerization by DIAPH1 is activated by RhoA. This activation of DIAPH1 is mediated by Cdk1 through phosphorylation of

Ect2, which in turn activates RhoA. Our results demonstrated that RhoA was immediately activated after release from RO-3306 treatment in HeLa cells whereas Cdk1-mediated phosphorylation of DIAPH1 reached a maximum at the metaphase to anaphase transition (Figure 1e). Thus, at the onset of mitotic rounding, DIAPH1 is first activated by Cdk1-Ect2-dependent RhoA activation. During mitotic progression, Cdk1-mediated phosphorylation of DIAPH1 occurs gradually and likely serves as a negative regulator of RhoA-activated DIAPH1, which consequently suppress actin polymerization at the cortex. This idea is further supported by the results showing that the actin polymerization activity of constitutively active DIAPH1 independent of RhoA is still inhibited by this phosphorylation (Figure 3e). The results are shown in the revised Figures 1e and 3e, and discussed in the text (page 8 bottom line-page 9 line 5, page 11 lines 6-10, and page 16 line 18-page 17 line 15).

ii) What is the mechanism by which cortical tension inhibits anaphase onset? If forces are communicated from the cortex to the spindle, one would expect cortical tension to aid kinetochore stretch. Explanations could include: the rigid cortex generated by activated DIAPH1 increases astral MT pushing forces, and/or DIAPH1 has a function within the spindle. Have the authors explored these possibilities?

This point is very important and interesting. We believe that the mechanism by which cortical tension regulates SAC is still an open question and requires further investigation. However, we also believe that a full elucidation of these mechanisms is likely beyond the scope of this manuscript at this stage. We discussed the above points in the revised Discussion section (page 18 line 19-page 19 line 10).

iii) HeLa cells have been reported to exit mitosis with normal kinetics when actin is depolymerized using latrunculin. Is this true for 3A cells?

According to the reviewer's suggestion, we examined the effect of latrunculin on the delay of anaphase onset in 3A cells and found that treatment with latrunculin reversed delay of the anaphase onset found in H2B-mCherry-KO-3A cells. The results are shown in the revised Supplementary Figure 6c and described in the text (page 14 lines 4-8).

2. Have the authors tested whether DIAPH1 activity in the activated mutant depends on Rho activity and/or Cdc42 (Oceguera-Yanez 2005, Yasuda et al., 2004, Rosa et al., 2015)?

As the reviewer indicated, we examined the effects of Rho inhibitor, C3 transferase, and the Cdc42 inhibitor, ZCL, on the enrichment of F-actin at the cortex in KO-3A cells and

found that only C3, but not ZCL, suppressed the enrichment of F-actin at the cortex in these cells. Thus, the results suggest that the actin polymerization activity of 3A mutant depends on Rho activity. The results are shown in the revised Figure 3f and discussed in the text (page 11 lines 10-12).

3. If, as the authors suggest, DIAPH1 phosphorylation reduces its association with Profilin, this should decrease actin filament assembly kinetics. It would be useful to look at cortical actin filament dynamics using FRAP.

According to the suggestion, we performed FRAP analysis using KO-Wt and KO-3A cells and found a rapid turnover of cortical F-actin in LifeAct-mCherry-KO-3A cells when compared to LifeAct-mCherry-KO-Wt cells, further supporting our idea that the actin polymerization activity of DIAPH1 is inhibited by its Cdk1-mediated phosphorylation. The results are shown in the revised Figure 3g and described in the text (page 11 line 20-page 12 line 5).

Does Profilin KD rescue the 3A actin phenotype? Also, do the DIAPH1 mutants affect actomyosin ring formation?

We also performed the knockdown of PFN1 in LifeAct-mCherry-KO-3A cells and found that PFN1 KD rescued the enhanced accumulation of F-actin at the cortex in these cells. The results are shown in the revised Figure 3f and described in the text (page line). We did not find any abnormality in actomyosin ring formation and subsequent cytokinesis in LifeAct-mCherry-KO-3A cells.

Specific points:

1. Figure 1A, there are no error bars. Also, the synchrony is very poor. Very few cells enter mitosis. Please show mitotic staging for 1B. It would also be good to state % anaphase in 1F.

According to the comment of the reviewer 1 (please see the above response), we eliminated the previous Figure 1a and showed the mitotic staging for the previous Figure 1b (the revised Figure 1a).

2. Figure 1H: it would be good to see immunofluorescence microscopy examples of cells in all phases of mitosis (including at early stages of mitotic rounding), and in the presence of the Cdk1 inhibitor (RO-Noc).

According to the reviewer's suggestion, we have included immunofluorescence microscopy examples of cells in all phases of mitosis in the revised Figure 1g and those in the presence

of the Cdk1 inhibitor in the revised Supplementary Figure 3c.

3. Figure 1D. What happened in lane 4?

This comment is the same as that of reviewer 1. Please see the above response.

4. Figure 3A, show DNA stain as internal intensity control. Please add scale bars.

We showed the DNA stain as an internal control in the revised Figure 3A. We put scale bars in all images of cells shown in the revised manuscript.

5. Figure 3C is very curious. It appears that cortical rigidity depends on time spent in metaphase. To test this, the plots should be blown up so one can directly compare the KO, KO-WT and KO-3A. It appears as if there is a steady increase in cortical contractility in the 3A cells during an extended mitosis. For comparison, one needs to see the same plot for WT cells that have been inhibited from exiting mitosis. The same is true for intensity – shown in 3A.

According to the reviewer's suggestion, we examined whether an extended duration of metaphase could affect the cortical rigidity and intensity of F-actin. We found that treatment of mitotic cells with MG132, which delays anaphase onset, failed to increase cortical rigidity measured by AFM or the intensity of F-actin. These results suggest that the increased cortical rigidity and intensity of F-actin is not simply due to an extended duration of metaphase, but likely to enhanced actin polymerization activity of 3A mutant. The results are shown in the revised Supplementary Figures 5a and 5b and described in the text (page 11 lines 14-19 and page 12 line 6-page 13 line 2).

6. Figure 3D: please change the labels of the x-axes of the upper and middle plot to 'Average force (pN)' and 'Maximum force (pN)' respectively .

According to the reviewer's suggestion, we changed the labels of the x-axes in the revised Figure 4c.

7. Figure 4A. The Figure should include rounding time.

As the reviewer indicated, we measured the rounding times of KD-Wt, -3A, and -3D cells and found them to be comparable. The results are shown in the revised Supplementary Figure 6a and described in the text (page 13 lines 15-17).

8. Figure 4F and G: It would be good to see how the intra-kinetochore distance and the percentage of 'stretching kinetochores' changes with tension, by decreasing tension in a

dose dependent manner with the MLCKi in KO-3A cells, since the authors made that point earlier with the anaphase onset and SAC positive cells measurements.

This point is important and well taken. We examined effect of MLCKi on the percentage of “stretching kinetochores”, and found that treatment with MLCKi reversed the reduction in the population of stretched kinetochores in KO-3A cells. The results are shown in the revised Figure 7c and described in the text (page 16 lines 12-15).

9. Please include confocal images of the spindle of the KO-3A cells, that were analysed for Extended Data Figure 4D.

According to the suggestion, we showed images of the spindle of KO-3A cells as well as those of KO-Wt and KO-3D cells in the revised Figure 6a and Supplementary Figure 7b, and described them in the text (page 15 lines 16-20).

Reviewer 3

1) The authors mention that “chromosome misalignment was not detected in KO-3A cells” but show no data to substantiate these claims. Considering the cortical tension has been linked to chromosome misalignments before (ref 21,22), and any defects here would delay mitotic exit (as shown in fig 4) it is critical that the authors properly quantify chromosome alignment (preferably with fixed analysis and live H2B movies).

This is very important issue and well taken. We analyzed chromosome alignment in H2B-mCherry-KO-Wt and -3A cells using both fixed and live H2B movie analyses, and found no remarkable chromosome misalignment in H3B-mCherry-KO-3A or -Wt cells. The results are shown in the revised Figures 6a and 6b, and Supplementary Movies 1 and 2) and discussed in the revised text (page 15 lines 12-20). In addition, we also failed to detect any changes in the cell diameter and spindle length in H3B-mCherry-KO-3A cells.

It is not clear how NEB to meta analysis in 4A and 4B was performed - where these in H2B tagged cells?

We measured the duration between NEB and metaphase, and that between metaphase to anaphase using H2B-mCherry-KO-Wt, -3A, and -3D cells. We added information on the cells used in the legends for Figures 5a and 5b, Supplementary Figures 6a, 6b, and 6c in the revised manuscript.

On a related note, it is surprising that reversine does not cause misalignment problems (i.e. increase NEBD-meta in b4) since this is well known to affect biorientation. Can the authors

explain this? This make me worry that the assays used were not sensitive enough to detect alignment differences, which has serious implications for conclusions about and the SAC (which could be an indirect consequence of a subtle alignment defect).

We optimized the concentration of reversine used in the experiments, so that no remarkable defects in chromosome biorientation were detectable in the live imaging analyses (reversine was used at a 500 nM concentration, which is less than that used in previous reports). We included this information in the revised manuscript (page 14 line 18- page 15 line 2).

2) It is not clear why pMCAK is chosen as a “SAC marker”. This is not a typical SAC marker and there are much better and more direct ones to use. I would suggest Mad2 as a marker a microtubule attachment and BUBR1 as a marker of tension. Both of these are well characterised SAC mediators that are needed for MCC assembly and are remove upon biorientation of kinetochore pairs. It would be important to quantify how Mad2/BubR1 change over time following microtubule attachment - is their loss from kinetochores delayed?

According to the reviewer’s suggestion, we examined the colocalization of BUBR1 and Mad2 with CREST, a centromere marker, and found that KO-3A cells showed a significant increase in the population of BUBR1-positive cells when compared with KO-Wt or KO-3D cells, whereas that of Mad2-positive cells appeared to be comparable in all these cells. These results suggest that the increased cortical tension in KO-3A cells likely causes impaired intra-kinetochore tension, but not chromosome misalignment. The results are shown in the revised Figures 5c and 5d and discussed in the text (page 14 lines 9-18).

3) The quantification of intrakinetochore stretch in 4f/g is appropriate, but I cannot understand the seemingly arbitrary cutoff of approx 0.16uM to defined a “stretched” kinetochore. Looking at the data this would imply that the vast majority of WT cells are not “stretched” at a time when the SAC is silenced. How can this be explained? It looks to me that a line is drawn to make a point out of slightly more outliers in the WT/3D population. It is not possible to tell from 4f what percentage of kinetochores in the WT situation are above this cutoff, but it looks like the vast a large minority which is why I refer to these points as “outliers”. Although there may be variations in stretch as the kinetochore pairs oscillate, the vast majority should still be in a stretched state at metaphase (i.e. above the noco mean). In this case I expect the differences in the 3A would be small and unlikely to explain the mitotic exit differences. It looks to me that there are no real difference in the level of intrakinetochore stretch, but that there may be subtle difference in a small pool of

hyperstretched kinetochores – would that be sufficient to explain the delayed exit. Do the authors conclude that a small amount of hyperstretched kinetochores are important to silence the SAC properly? I fail to see how these data substantiates the authors model.

Kinetochores stretching is a phenomenon characterized by repetitive cycles of extension and recoiling of the kinetochore structure which emerges upon microtubule attachment. One experiment to estimate the degree of stretching is to quantitate how many kinetochores are detectably extended above a threshold level, which is defined by taking a 95% confidential interval of the intra-kinetochore distance in the absence of microtubule attachments, the condition created by nocodazole treatment (Uchida et al., 2009).

In the current experimental setup, a statistical calculation based on the data obtained with nocodazole treatment gave a threshold value of 0.16 μm . This allowed us to estimate that, in Wt cells, $\sim 15\%$ of kinetochores are extended and deformation of the remaining $\sim 85\%$ was below the threshold level, at any given time in unperturbed metaphase. Importantly, this does not mean that only $\sim 15\%$ of kinetochores are statically stretched and the rest unstretched, but it does indicate that a majority of cellular kinetochores undergo a back and forth switching between extension and recoiling states, which allows inactivation of the spindle-assembly checkpoint. A decrease in this rate has been shown to perturb SAC inactivation and to delay anaphase onset (Uchida et al., 2009). Therefore, a decrease in the stretching rate to $\sim 7\%$ in the 3A mutant must be significant (5% is the minimum value, by definition) and thus sufficiently explains the mitotic delay.

There 3 major caveats above mean that I am not convinced that the clear mechanistic data actually causes a defect in SAC silencing. Another key experiment that could help to change that (in addition to the ones above): Cyclin B reporters to accurately quantify when Cyclin B degradation starts (i.e. a true marker of SAC inactivation) in relation to chromosome alignment, or better still, Mad2/BubR1 loss from the last KT.

This point is very important and well taken. We examined cyclin B degradation using GFP-cyclin B-KO-Wt and -3A cells during mitosis and found that reduction in the GFP-cyclin B intensity after metaphase was significantly delayed in GFP-cyclin B-KO-3A cells compared to that in GFP-cyclin B-KO-Wt or GFP-cyclin B-KO-3D cells. Thus, the results suggest that Cdk-mediated phosphorylation of DIAPH1 regulates inactivation of SAC at an appropriate time through suppression of excess accumulation of F-actin at the cortex. The results are shown in the revised Figure 5e and discussed in the text (page 15 lines 5-11).

In addition to the points above I have one major point regarding statistics. It is important that the authors clearly state the n numbers and what these refer to in the legends. This is

critical because, for example, figure 4a and 4b show the critical increase in meta to ana duration in single cells, but it is unclear how many experiments this is from.⁹ It looks like the p values are calculated as if n = number of cells, which is not correct at all - a mean meta-ana duration for each experiment should be performed (i.e. means of “X” number of cells) then the stats calculated from those means based on n = number of experiments (for clarification see: Lazic et al, Plos Biol 2018). This should be clarified in the legends and applied to all figures panel throughout. Currently many of the panel do not state the n number.

This is a very important point. We added the n number in all legends. With respect to the p values and statistical analysis, according to the reviewer’s indication and clarification by Lazic et al., we deleted all p values from the revised Figures whose results were obtained from one set of the experiment.

Other minor points:

P9, L151 “decreasing” population of

We changed the wording.

REVIEWERS' COMMENTS:

Reviewer #1 (Remarks to the Author):

The authors have done a good job in improving their manuscript.

Minor points remaining are :

- figure 6 is supplementary information.
- figure 7 requires statistical analysis for quantifications shown (e.g. 7c), which in fact should be performed as independent set of experiments rather than a quantisation of 5 cells!

Reviewer #2 (Remarks to the Author):

The revised paper contains many novel findings that will interest a wide audience. There are a few outstanding questions that the authors should consider addressing. However, overall it is a thorough and impressive piece of work.

Figure 1e:

Is the increase in pDIAPH1 Ect2/RhoA dependent or not?
It would be good to show timing relative to pERM/pMyosin.

Figure 3a:

The actin images are of very poor quality.
It is important to show better images and to determine whether there is an increase in cytoplasmic actin following active Diaph1-3A expression. This is especially the case, given the localisation of Diaph1 in images in the paper.

Figure 3g. It is important to show images of the cells before, during and after FRAP.
How have the authors dealt with changes in the FRAP recovery that are the result of differences in signal/background ?

Figure 4a/b.

The paper makes the case that Diaph1 is not phosphorylated at the onset of rounding - only later during metaphase. Why then are the graphs in 4a and 4b so different?
The authors should show a plot of the initial upward force in the first minutes of rounding - based on the work in Figure 1, this would be expected to be the same for both 3A and control cells. Is this the case? If not, why not?

Figure 7. The individual data points are not clear as shown.
Where are the statistics for 7b and c?

Finally, the authors state that there is no impact on cytokinesis.
It would be good to show data to support this and to measure the rate of furrow closure/polar relaxation..
If Diaph1 is hyperactive - one would expect this to have an impact anaphase shape changes.

Reviewer #3 (Remarks to the Author):

The authors have made a good effort to address my main concerns. I would just add two comments

about points 1 and 3.

1) the authors need to add quantifications for this. One example image does not say anything, and to have a whole figure 6 dedicated to an example image is not appropriate. I suggest quantifying alignment time and any defective anaphases (i.e. lagging chromosomes or unaligned chromosomes). I presume the authors have many movies that could easily be quantified for this purpose given. This figure could also easily be place in the supplements instead.

3) I accept the authors rationale here, but this needs to be clearly stated in the methods/results (if it is not already).

Point-by-point responses to the reviewers' comments

Reviewer 1

-figure 6 is supplementary information.

According to the reviewer's suggestion, we put the previous figure 6 as a revised supplementary figure 8b and 8c.

- figure 7 requires statistical analysis for quantifications shown (e.g. 7c), which in fact should be performed as independent set of experiments rather than a quantisation of 5 cells!

This point is important. Although the statistical analysis requires the results from an independent set of experiments, we have already analyzed more than 200 kinetochores ($n > 200$) in each cell types (4 or 5 independent cells). Therefore, we believe that our results showing a clear difference in the percentage of stretching kinetochores between DIAPH1-KO-Wt and DIAPH1-KO-3A cells are experimentally meaningful. In place, in order to make these points more clear for the readers, we showed the individual data points as a box-and-whisker plot in the revised Figure 6c (previous Figure 7c).

Reviewer 2

Figure 1e: Is the increase in pDIAPH1 Ect2/RhoA dependent or not?

It would be good to show timing relative to pERM/pMyosin.

According to the reviewer's suggestion, we examined whether phosphorylation of DIAPH1 required Ect2/RhoA activation and found that it did not. In addition, we also determined the timing of pMLC2. These results are shown in the revised supplementary figure 3c and figure 3e, respectively and discussed in the text (page 8 bottom line and page 9 lines 3-5).

Figure 3a: The actin images are of very poor quality.

It is important to show better images and to determine whether there is an increase in cytoplasmic actin following active Diaph1-3A expression. This is especially the case, given the localisation of Diaph1 in images in the paper.

This point is important and well taken. We showed the better actin images in the revised Figure 3a. DIAPH1-3A is not a constitutive active form but an uninhibitable form (actin polymerization activity of DIAPH1-3A is not inhibited by Cdk-mediated phosphorylation). The DIAPH1-3A activity still requires activated RhoA which predominantly localizes at the cortex. Therefore, we failed to detect an increase in cytoplasmic actin polymerization in DIAPH1-3A cells.

Figure 3g. It is important to show images of the cells before, during and after FRAP.

How have the authors dealt with changes in the FRAP recovery that are the result of differences in signal/background ?

This point is also important and well taken. We put images of cells before, during and after FRAP analysis in the revised supplementary figure 6. We also added the detailed information about how to calculate the FRAP recovery in the revised legend for figure 3g.

Figure 4a/b.

The paper makes the case that Diaph1 is not phosphorylated at the onset of rounding - only later during metaphase. Why then are the graphs in 4a and 4b so different?

The authors should show a plot of the initial upward force in the first minutes of rounding - based on the work in Figure 1, this would be expected to be the same for both 3A and control cells. Is this the case?

In these experiments, we measured the upward force of cells just after rounding (not before rounding). Therefore, we had difficulty in measuring the initial upward force by our AFM.

Figure 7. The individual data points are not clear as shown.

Where are the statistics for 7b and c?

We showed the individual data points as a box-and-whisker plot in the revised Figure 6c (previous Figure 7c). With respect to the statistics, please see the above response (comment from the reviewer 1).

Finally, the authors state that there is no impact on cytokinesis.

It would be good to show data to support this and to measure the rate of furrow closure/polar relaxation.

According to the reviewer's suggestion, we examined the rate of cells showing abnormal cytokinesis in DIAPH1-KO-Wt and DIAPH1-KO-3A cells and found that their rates appeared comparable in these cells. The results are shown in the revised supplementary figure 8d and discussed in the text (page 15 lines 17-18).

If Diaph1 is hyperactive - one would expect this to have an impact anaphase shape changes. Please see the above response (2nd comment from the reviewer).

Reviewer 3

1) the authors need to add quantifications for this. One example image does not say

anything, and to have a whole figure 6 dedicated to an example image is not appropriate. I suggest quantifying alignment time and any defective anaphases (i.e. lagging chromosomes or unaligned chromosomes). I presume the authors have many movies that could easily be quantified for this purpose given. This figure could also easily be placed in the supplements instead.

According to the suggestion, we quantified the defective anaphases by counting cells with lagging chromosomes and found that cells with lagging chromosomes were comparable between DIAPH1-KO-Wt and DIAPH1-KO-3A cells. The results are shown in the revised supplementary figure 8d and discussed in the text (page 15 lines 15-20). With respect to the alignment time, we have already measured the duration between NEBD and Meta phase, and Meta phase and Ana phase in DIAPH1-KO-Wt and DIAPH1-KO-3A cells. The results were shown in the figure 5a and 5b.

3) I accept the authors rationale here, but this needs to be clearly stated in the methods/results (if it is not already).

According to the suggestion, we added the rationale behind our calculation of kinetochore stretching in the revised Methods section.